# Efficacy and Tolerance of Antipsychotics Used for the Treatment of Patients Newly Diagnosed with Schizophrenia: A Systematic Review and Meta-Analysis

**DOI:** 10.3390/pharmacy11060175

**Published:** 2023-11-10

**Authors:** Zina Sherzad Qadir, Patrick Anthony Ball, Hana Morrissey

**Affiliations:** 1Research Institute in Health Sciences, University of Wolverhampton, Wolverhampton WV1 1LY, UK; sherzad.zhina@gmail.com (Z.S.Q.);; 2School of Dentistry & Medical Sciences, Charles Sturt University, North Wagga, NSW 2650, Australia

**Keywords:** antipsychotics, antipsychotics discontinuation rate, causes of relapse, optimisation of schizophrenia management

## Abstract

This systematic review compared the efficacy and tolerance of oral antipsychotics (APDs) used in the treatment of schizophrenia following the PRISMA-P© statement (*n* = 21). The primary outcomes of interest were clinical response measured with symptoms’ improvement, tolerance to side effects and discontinuation reasons. There was better individual patients’ response to aripiprazole vs. ziprasidone and quetiapine ((CDSS *p* = 0.04), BPRS *p* = 0.02, YMRS *p* = 0.001) and ziprasidone vs. quetiapine (CGI *p* = 0.02, CDSS *p* = 0.02). Aripiprazole was more tolerated than risperidone, ziprasidone and quetiapine (*p* < 0.05). Quetiapine was more tolerated than aripiprazole, ziprasidone and risperidone (*p* < 0.05). Ziprasidone was more tolerated than quetiapine haloperidol and olanzapine (*p* < 0.05). Risperidone was more tolerated than olanzapine (*p* = 0.03) and haloperidol was more tolerated than olanzapine and quetiapine (*p* < 0.05). Olanzapine caused less discontinuation than quetiapine; quetiapine caused less discontinuation than ziprasidone, aripiprazole and haloperidol; ziprasidone caused less discontinuation than quetiapine, aripiprazole and haloperidol; aripiprazole caused less discontinuation than quetiapine, ziprasidone and olanzapine and olanzapine caused less discontinuation than ziprasidone and haloperidol (*p* < 0.05). It was concluded that individual patient clinical response, tolerance to side effects and life-threatening side effects remain the most reliable basis for selecting and continuing the use of APD.

## 1. Introduction

Schizophrenia affects 24 million (1 in 300, 0.32%) people worldwide and men are more affected and generally at a younger age than women [1]. Acute, uncontrolled schizophrenia can lead to disability, impairment and severe distress. An individual diagnosed with schizophrenia has a higher likelihood (2–3 fold) of dying prematurely compared to the general population, due to infectious diseases, metabolic disorders and cardiovascular issues [1]. The symptoms of schizophrenia are classed into positive symptoms (hallucination, delusion and disordered thinking) and negative symptoms (social withdrawal, appear emotionless and flat, disorganised speech, lack of drive and self-neglect) [2].

In England, the annual cost of schizophrenia to society and the public sector is GBP 11.8 billion and GBP 7.2 billion, respectively [3]. The increased mortality risk in people with schizophrenia is complex. The incidence of psychosis in schizophrenia triggers a cascade of socioeconomic and lifestyle factors, including unhealthy lifestyle behaviour such as excessive alcohol consumption, smoking and poor diet that in return can result in adverse physical health outcomes [4,5]. Suicide is the most common cause of death in this population; however, individuals with schizophrenia have been found to experience increased mortality from all causes [6]. Semahegn et al. [7] found that 49% of patients diagnosed with schizophrenia did not adhere to therapy, and this was higher than in patients diagnosed with major depressive disorders, and bipolar disorders (50% and 44%, respectively). Relapse risk remains high after the first episode of psychosis, following discontinuation of initial antipsychotic therapy, which indicate that in some patients, ongoing antipsychotic prophylaxis is required after recovery from a first psychotic episode [8]. Boonstra et al. [8] concluded that antipsychotic prophylaxis regimens play a key role in maintaining remission, and a higher risk of relapse was associated with their discontinuation than with their continuation (*p* = 0.001) after 9 months from the first episode of schizophrenia. Similarly, Leucht et al. [9] reported that when antipsychotic medications were compared to a placebo, they were shown to significantly reduce the relapse rate (27% vs. 64%; [RR] 0·40, 95% CI: 0.33–0.49) and involve fewer hospital admissions (10% vs. 26%; RR: 0.38, 95% CI: 0.27–0.55), but there was limited evidence for quality of life improvement (SMD:.62, 95% CI: 1.15 to 0.09) in patients with schizophrenia. Patients who experienced relapse were characterised by higher rates of hospitalisation and their cost of treatment averaged as over four times higher than a control group who did not experience relapse for the last 6 months [10].

The choice of an antipsychotic medication should consider the severity of the side effects such as extrapyramidal symptoms (EPSs), cardiovascular symptoms, metabolic symptoms and hormonal imbalance [11]. The annual prevalence of antipsychotic medication use is on the rise; usage in the UK increased by 29.3% from 2005 to 2012 [12]. Prolactin-sparing antipsychotic agents such as olanzapine, quetiapine, aripiprazole and clozapine are considered less likely to cause sexual side effects such as decreased libido, impaired arousal and impaired orgasm [13]. Meanwhile, risperidone, amisulpride and sulpiride and first-generation antipsychotics are associated with hyperprolactinaemia [11]. Moreover, reducing cholinergic receptors and alpha-adrenergic alpha receptors reduces peripheral vasodilation that could lead to erectile dysfunction. Also, histamine antagonism can impair arousal by directly increasing sedation [14].

The positive and negative symptoms scale (PANSS) and clinical global impression scale (CGI) are rating systems to provide a comprehensive measure of symptomatology, consisting of 18 items from the brief psychiatric rating scale (BPRS) to measure positive, negative and general psychopathology and indicate general symptom severity [15,16]. The CGI provides an overall index of symptom severity or change, which makes it more practical to administer, while the PANSS is more comprehensive to use. Both PANSS and CGI show a high degree of correspondence. In addition, reduction in the BPRS/PANSS corresponds to reduction in CGI severity. However, PANSS mostly assesses specific symptoms (e.g., positive and negative); therefore, they are not interchangeable, and it is beneficial to use both measures [15,17,18] Table 1 provides further explanation about all scales used in this systematic review analysis and discussion.

## 2. Review Aim

The aim of this review was to compare the evidence of effectiveness and tolerability of antipsychotic drugs (APDs) to treat psychosis in APD-naïve (first episode of psychosis) patients after both short-term (≤12 weeks) and long-term (>12 weeks) use. The following aspects were investigated, based upon evidence published in the current literature:Comparing the effectiveness of APDs used for the treatment of adults diagnosed with schizophrenia.Comparing the tolerability of APDs used for the adult patients diagnosed with schizophrenia.Comparing the discontinuation rate and reason of APDs used for the adult patients diagnosed with schizophrenia.

## 3. Methods and Design

This systematic review was reported according to the PRISMA statement and used a PRISMA flow chart to depict the process of the search strategy for the included published papers, following the Cochrane Handbook for Systematic reviewer’s methodological guidelines [19]. This project was prospectively registered and submitted on the PROSPERO database (registration number: CRD42022311060, appendix 4). Data synthesis was performed using Review Manager (RevMan©) software V.5.4.1. A statistical analysis was conducted whenever similar data including continuous and dichotomous outcomes were available, and, in cases when it was not possible, a narrative analysis was conducted.

For both continuous and dichotomous outcomes, the mean difference and relative risk (RR), respectively, with a 95% confidence interval (CI) were reported. Results were considered significant when *p* < 0.05 and when the upper limit of 95% CI was less than 1 and the lower limit did not cross the line of no effect. A total effect estimate for dichotomous data was calculated as RR with 95% CI. In dichotomous data, the pooled effect size with 95% CI was obtained using a random effect (RE) model and fixed effect (FE) model. The RR with 95% CI was calculated using the Cochran–Mantel–Haenszel (M-H2) test.

An overall effect estimate for a continuous outcome was reported as the mean difference (MD). The effect sizes for continuous outcomes were centred on zero MD, values greater than zero favoured the intervention group and those less than zero were favouring the comparisons. For studies reporting a continuous outcome, effect size was calculated using the inverse variance (IV) method. *I*^2^ describes the percentage of total variation across trials that is due to heterogeneity rather than chance or sampling errors [20]. When the *I*^2^ value was 30% or under for subgroups, the FE model was used for reporting, and when it was >30%, the RE model was used for more precision of reporting on the results.

A subgroup analysis was performed where there was more than one variable or indicators to demonstrate outcomes and when sufficient data were available (more than one study). The subgroup analysis determined the difference in pooled effect sizes between subgroups [21]. Subdividing data into subgroups was performed to explore diverse outcomes, or to address particular inquiries concerning specific categories of interventions. As the systematic review was time-limited, publications from January 2000 to December 2021 were considered suitable for inclusion. The first APD was in use since 1950, the newest was marketed in 2020 and the newest FDA was registered in 2021. The systematic review ended in January 2022 and only randomised clinical trials (RCTs), which included oral routes of administration (tablets), were included.

## 4. Search Strategy

The PICO framework (population, intervention, control comparison intervention treatment/placebo/standard of care, outcomes) was used where the following applied:

Population (P): 15 years of age or over, APD-naïve patients, diagnosed with confirmed schizophrenia or presenting psychosis with short-term (0–12 weeks) or long-term (>12 weeks) treatment.

Intervention (I): Treatment with APD.

Comparators (C): Head to head, one APD compared to another APD.

Outcomes (O): The outcome will determine the clinical effectiveness of the APDs in managing the symptoms, remission, preventing relapse and emerging of a side effect and promoting adherence to therapy.

Settings (S): Community, primary or secondary care.

Three databases were reviewed: PubMed©, CINHAL© and ScienceDirect™. The medical subject headings (MeSH) search terms used in the PubMed database were (Randomised OR randomized) AND (schizophrenia) AND (naïve OR first-episode) AND (efficacy OR effectiveness OR effect) AND (adherence OR non-adherence OR compliance OR discontinuation OR withdrawal OR remission) AND (antipsychotic OR psychotropic OR psychotic medication OR psychiatric medication) AND (complication OR hospitalization OR relapse OR side effect OR tolerability OR adverse effect OR symptom OR risk OR clinical).

## 5. Selection Criteria

-Adults diagnosed with schizophrenia aged 15 years old or over.-APD vs. APD/s, independent of whether they were first-generation (FGA) or second-generation (SGA) agents or the dose of administration.-APD-naïve patients or lifetime APD short-term use history.

A total of 14,417 articles were identified; the search was then restricted to the period of 2020–2021, and ‘oral antipsychotics vs. oral antipsychotics’ was added, which reduced the total publications to 166. The titles and abstracts were then screened to identify studies that meet the inclusion criteria, where 145 were removed as shown in Figure 1, leaving 21 publications for inclusion in the systematic review. The authors individually screened all studies and in case of disagreement, it was resolved with consensus (Appendix A).

## 6. Data Analysis

The analysis model (FE vs. RE) was selected based on heterogeneity, when the *I*^2^ value was above 30% for any of the subgroups; reporting was based on the RE model for precision of findings [22]. The forest plots were only possible when there were two or more studies that shared similar outcomes. The statistical unit for any of the domains of study was based on numbers of patients. Due to lack of numbers of studies in each domain of the analysis (less than 10 studies), a funnel plot was not produced to assess publication bias [23]. In addition, the chi-squared (X^2^, or Chi^2^) test was used for heterogeneity in the forest plots using the formulae of *I*^2^ = 100 × X^2^f/X^2^ for quantifying the inconsistency of the discontinuation rate in one APD vs. another.

The included studies data reporting was inconsistent, as some studies presented the results in multiple means, e.g., means, standard means, mean difference, standard division, mean standard division, range, points or score, frequencies or percentages; accordingly, it was not possible to collate or further analyse. When a statistical analysis was not possible, reported results were tabulated and narratively explained.

## 7. Results

### 7.1. Symptoms’ Improvement (Efficacy)

#### 7.1.1. Efficacy of Antipsychotics after Short-Term Treatment

Comparing PANSS, BPRS, CGI and CDSS, no significant differences were reported (*p* > 0.05) [24,25,26]. For olanzapine and haloperidol, PANSS (total *p* = 0.02 and *p* = 0.019, negative *p* = 004 and general *p* = 0.003) and MADR (*p* = 0.02) were significantly different between the two APDs [25,27]. For olanzapine and quetiapine, only PANSS negative scores were significantly different (*p* = 0.017) in the study by McEvoy et al. [28] but not (*p* > 0.05) in the study by San et al. [25]. Comparing olanzapine and risperidone, it was concluded that there were no significant differences between the two APDs (*p* > 0.05) [25,26,28]. Only PANSS positive symptoms’ scores were significantly different (*p* = 0.031); other reported scores were not significantly different between quetiapine and risperidone (*p* > 0.05) [25,26,28].

Quetiapine and haloperidol were compared at the end of 12 weeks, and except for GAF (*p* = 0.79) and PANS (*p* > 0.05) total scores, all other scales’ scores were significant between quetiapine and haloperidol (*p* < 0.05) [25,29]. Regarding ziprasidone and quetiapine at the end of week 12, scores’ reduction in terms of quetiapine (0.50 ± 0.11) was slightly less on BPRS than ziprasidone (0.54 ± 0.13) (*p* > 0.05) and a similar rate of BPRS total score reduction was seen for risperidone (0.55 ± 0.12) and ziprasidone (0.54 ± 0.13) (*p* > 0.05) [25,26]. Wang et al. [26] found that the BPRS total score for olanzapine (0.47 ± 0.15) was slightly higher than aripiprazole (0.44 ± 0.13). However, aripiprazole (0.44 ± 0.13) appeared to be less efficacious than ziprasidone (0.54 ± 0.13) in the BPRS total score, but the difference was not significant (*p* > 0.05). Comparing risperidone and aripiprazole, it was reported that only BPRS and the ‘disorganised’ dimension showed a statistically significant difference between the two APDs (*p* < 0.001) [26,30,31]. Additionally, Robinson et al. [31] reported CGI, asociality–anhedonia, alogia and affective flattening were significantly different at *p* < 0.001, as well as SANS avolition–apathy global scores (*p* = 0.03). At 12 weeks, Lieberman et al. [32] found that only the SANS total score was significant between clozapine and chlorpromazine (*p* = 0.01).

#### 7.1.2. Efficacy of Antipsychotics after Long-Term Treatment

The mean score at the endpoint analysis for aripiprazole and ziprasidone showed that the heterogeneity between the included studies was considerable in subgroup 88.2.5 (*I*^2^ = 81%); the RE model was also used to ensure high precision of the reporting on this outcome. The mean change from baseline to the endpoint analysis showed that the heterogeneity on the FE model between the included studies was low (*I*^2^ = 0–3%) for the subgroups. There was no significant difference in any of the subgroups’ analyses except for the CDSS mean score at the endpoint (*p* = 0.04) and BPRS mean change between the baseline and endpoint (*p* < 0.001). Ziprasidone showed a lower mean score at the endpoint on CGI, BPRS, SANS and CDSS and a higher mean change from the baseline to endpoint on BPRS, SANS and CDSS. Aripiprazole showed a lower mean score at the endpoint on SAPS and YMRS and a higher mean change from the baseline to endpoint on SAPS and YMRS (Table 2).

The mean score at the endpoint analysis for aripiprazole and quetiapine showed that the heterogeneity between the included studies was moderate in subgroup 88.1.5 (*I*^2^ = 52%). Accordingly, the RE model was used to ensure high precision of the reporting on this outcome. Additionally, the mean change from the baseline to endpoint analysis showed that the heterogeneity on the FE model between the included studies was low (*I*^2^ = 0–6%) for all the subgroups. There were no significant differences in any of the subgroups’ analyses except for BPRS mean change between the baseline and endpoint (*p* = 0.002) and the YMRS mean score at the endpoint (*p* = 0.01). Regarding symptoms’ improvement, aripiprazole showed a lower mean score at the endpoint on CGI, BPRS, SAPS and YMRS and a higher mean change between the baseline and endpoint on all scales except CDSS. Quetiapine showed a lower mean score at the endpoint and a higher mean change between the baseline and endpoint on CDSS only (Table 3).

The mean score at the endpoint analysis for ziprasidone and quetiapine showed that the heterogeneity between the included studies was low in all subgroups. Additionally, the mean change from the baseline to endpoint analysis showed that the heterogeneity on the FE model between the included studies was low (*I*^2^ = 0%) for all the subgroups. There was no significant difference in any of the subgroups’ analyses except for the CGI (*p* = 0.02) and CDSS (*p* = 0.020) mean score at the endpoint. Regarding symptoms’ improvement, ziprasidone showed a lower mean score at the endpoint on CGI, BPRS, SANS, SAPS and YMRS and a higher mean change between the baseline and endpoint on CGI, SANS and SAPS. Quetiapine showed a lower mean score at the endpoint on CDSS only and a higher mean change from the baseline to endpoint on BPRS, SAPS, CDSS and YMRS (Table 4).

For olanzapine and haloperidol, significant differences were reported for remission (*p* = 0.036) and MADRS (*p* = 0.045) and PANSS total scores (*p* < 0.001) [25,36]; however, there were no significant differences between olanzapine and risperidone reported (*p* > 0.05) [25,28,37]. For olanzapine and quetiapine, the ‘disorganized’ mean change at 3 years [37], total PANSS [38] and positive PANSS at 52 weeks [28] were the only significantly different parameters (*p* < 0.05, *p* < 0.001, *p* = 0.013, respectively). For olanzapine and ziprasidone, CGI total mean change and SAPS mean change at 3 years [36] and PANSS total reduction at 12 months [36] were the only significant parameters between the two drugs (*p* < 0.05, *p* < 0.05, *p* < 0.001, respectively). Regarding ziprasidone and haloperidol, at 3 years, CGI total mean change, CDSS mean change, positive mean change [36] and PANSS total reduction [38] at 12 months showed significant differences between the two drugs (*p* < 0.05, *p* < 0.05, *p* < 0.05, *p* < 0.001, respectively). For quetiapine and haloperidol, at 3 years, CDSS mean change [35] and total PANSS [36] were statistically significantly different (*p* < 0.05, *p* < 0.001, respectively). The chlorpromazine group took significantly longer to achieve remission compared to the clozapine group [32,38]. Kahn et al. [37] showed that the PANSS score significantly decreased for amisulpride and olanzapine compared to all other APDs (*p* < 0.001) between the baseline and the 12-month reviews, but there were no significant differences between all other APDs (*p* > 0.05).

### 7.2. Side Effects

#### 7.2.1. Tolerance of Antipsychotics after Short-Term Treatment

Cardiovascular side effects were not intentionally excluded, as the data identified in the included studies were insufficient to focus on cardiovascular outcomes in particular. The analysis included all side effects that were reported in the studies included in this review. Included studies reported on all or some of the following: concentration difficulties, increased fatigability, sleepiness, memory impairment, depression, restlessness, increased duration of sleep, rigidity, akinesia, tremors, increased salivation, constipation, vertigo, amenorrhea, galactorrhoea, diminished sexual desire, orgasmic dysfunction, erectile dysfunction, ejaculatory dysfunction and weight gain.

The heterogeneity between Gómez-Revuelta et al. [30] and Robinson et al.’s [31] studies was moderate in subgroup 42.1.1 (*I*^2^ = 60%), and the RE model analysis was performed. The diminished sexual desire subgroup was the only subgroup with a significant difference between the two APDs, favouring aripiprazole (4.7%) over risperidone (12.5%) (*p* = 0.01) (Table 5).

For akathisia after short-term treatment, the heterogeneity of the included studies was moderate (*I*^2^ = 32%) in the FE model; RE was created and the finding was reported based on the RE (Figure 2). There was no significant difference between aripiprazole and risperidone (*p* = 0.440); however, akathisia events were higher with aripiprazole (18.7%) compared to risperidone (15.2%).

Robinson et al. [32] suggested that parkinsonian symptoms’ prevalence was lower with aripiprazole (14.8%) than risperidone (15.5%), but not significantly different (*p* = 0.750). For cumulative EPS, the heterogeneity of the included studies was low (*I*^2^ = 0%) and the FE was used to report on this outcome (Figure 3). There was no significant difference (*p* = 0.25) but EPS events were higher with aripiprazole (13%) than risperidone (9.5%). The reported RR (95% CI) was 1.22 (0.75, 3.06), indicating that the type of the drug had a high impact on causing treatment-emerged EPS in the study population.

Wang et al. [26] and Gómez-Revuelta et al. [30] reported that all side effects for aripiprazole and risperidone were significantly different (*p* < 0.05) except for leukopaenia and ECG abnormalities at week 6. Amr et al. [29] and San et al. [25] reported on side effects for quetiapine and haloperidol, and only weight gain, blood glucose levels, insomnia and dizziness were not significantly different (*p* > 0.05). San et al. [25], Crespo-Facorro et al. [39] and Wang et al. [26] reported on side effects for quetiapine and ziprasidone, where weight gain, increased duration of sleep and somnolence were significantly different (*p* = 0.003 for all).

Crespo-Facorro et al. [40] and Wang et al. [26] found that weight gain, somnolence, increased duration of sleep, treatment-emergent akathisia and EPS were significantly different between quetiapine and aripiprazole (*p* < 0.05). San et al. [25] and Wang et al. [26] reported only UKU neurological symptoms, EPS and total UKU were significantly different between olanzapine and risperidone (*p* < 0.05).

San et al. [25] reported that weight gain and the glucose level were greater for risperidone (+8 kg, 4.7 ± 0.5) than haloperidol (+4 kg, 4.5 ± 0.4) but the difference was not significant (*p* > 0.05). UKU total score change was greater with haloperidol (11.6 ± 8.8, +9.5) than risperidone (9 ± 6.7, +6.8) (*p* > 0.05). Two studies [25,26] reported only parkinsonian symptoms (*p* < 0.05) were significantly different between quetiapine and risperidone (*p* > 0.05). Grootens et al. [24] reported that weight gain, mean weight gain, blood glucose levels, triglycerides, cholesterol and liver transaminases were significantly different (*p* < 0.05) between ziprasidone and olanzapine. San et al. [25] and Wang et al. [26] reported that only UKU psychiatric symptoms were significantly different (*p* < 0.05).

Two studies [26,39] reported that weight gain (*p* = 0.003), abnormal ECG (*p* < 0.05) and EPS (*p* < 0.05) were significantly different between ziprasidone and aripiprazole. San et al. [25] reported that only UKU neurological symptoms (*p* = 0.033) and total UKU (*p* = 0.008) were significantly different between haloperidol and ziprasidone. Wang et al. [26] showed a significantly higher rate of EPS with aripiprazole compared to olanzapine (40% vs. 5%, *p* < 0.05).

Tolerance of antipsychotics after long-term treatment

The FE model showed low heterogeneity between studies comparing aripiprazole and ziprasidone in all subgroups’ analyses (*I*^2^ between 0 and 30%). Table 6 illustrates that there was a significant difference between the two drugs in six subgroups: sleepiness (*p* = 0.03), increased duration of sleep (*p* = 0.003), rigidity (*p* = 0.02), erectile dysfunction (*p* = 0.005), ejaculatory dysfunction (*p* = 0.02) and weight gain (*p* = 0.01). Based on the reported 95% CI, the type of APDs had no consistent impact on causing side effects in the study population.

The heterogeneity of the included studies was low (*I*^2^ = 1%) and the FE model was used to report this outcome. There was no significant difference between the two drugs (*p* = 0.28), where akathisia events were higher with ziprasidone (31%) than aripiprazole (25.5%). The reported RR (95% CI) was 0.83 (0.60, 1.16), indicating that the type of the drug had a low impact on causing treatment-emerged akathisia in the study population (Figure 4).

Crespo-Facorro et al. [34] concluded that parkinsonian symptoms were higher with ziprasidone (19.6%) than aripiprazole (17.7%) (*p* > 0.05). The heterogeneity of the included studies was low (*I*^2^ = 0%). The EPS total events were not significantly different (*p* = 0.85); however, events were higher with aripiprazole (21.2%) than ziprasidone (20.2%). The reported RR (95% CI) was 1.05 (0.64, 1.74), indicating that the type of the drug had no impact on EPS development (Figure 5).

The FE model showed low heterogeneity between all studies in the subgroups’ analyses (*I*^2^ = 0%) but it was moderate in subgroup 15.1.1 (*I*^2^ = 32%) and substantial in subgroup 15.1.17 (*I*^2^ = 69%); the RE analysis was used for reporting on this outcome. The side effect profile was significantly different in five subgroups: sleepiness (*p* < 0.001), increased duration of sleep (*p* = 0.001), tremors (*p* = 0.04), erectile dysfunction (*p* = 0.002) and galactorrhoea (*p* = 0.03). Based on the reported RR, the type of APDs had a low impact on causing side effects in the study population (Table 7).

Crespo-Facorro et al. [34] reported that hyperprolactinemia was less with aripiprazole (19.6%) than quetiapine (44.4%) (*p* < 0.05), but parkinsonian symptoms (17.7% vs. 14.3%) were not significantly different (*p* > 0.05). The heterogeneity of the included studies was low (*I*^2^ = 0%); the FE model was used to report on this akathisia (Figure 6), which was significantly different (*p* = 0.05), with higher prevalence with aripiprazole (25.5%) than quetiapine (16.3%). The reported RR (95% CI) was 1.57 (1.01, 2.44), indicating that the type of the drug had a high impact on causing treatment-emerged akathisia in the study population.

Heterogeneity between all studies comparing quetiapine vs. ziprasidone was low for most of the subgroups’ analyses (*I*^2^ = 0–18%) but it was moderate in subgroups 16.1.10 (*I*^2^ = 53%), 16.1.17 (*I*^2^ = 36%) and 16.1.18 (*I*^2^ = 44%); the RE analysis was used for reporting on this outcome. There were significant differences in four subgroups: rigidity (*p* < 0.03), vertigo (*p* = 0.05), amenorrhoea (*p* = 0.006) and weight gain (*p* = 0.003). Based on the reported RR, the type of APDs had a low impact on causing side effects in the study population (Table 8).

The heterogeneity of the three included studies was low (*I*^2^ = 0%); the FE model was used to report on akathisia (Figure 7). There was a significant difference between the two drugs (*p* = 0.005); akathisia events were higher with ziprasidone (30.6%) than quetiapine (16.3%). The reported RR (95% CI) was >1 (1.87 [1.21, 2.91]), indicating that the type of the drug had a high impact on causing treatment-emerged akathisia in the study population.

At week 52, San et al. [25] did not observe significant differences (*p* > 0.05) between quetiapine and olanzapine in UKU psychiatric side effects (2.3 ± 1.4 vs. 1.7 ± 1.8, respectively), neurological side effects (0.3 ± 0.8 vs. 0 ± 0, respectively) or the glucose level that showed to be similar for both drugs (4.6 ± 0.5 vs. 4.6 ± 0.7, respectively). Weight gain was slightly higher with olanzapine (+9 Kg) than quetiapine (+6 Kg) but remained not significantly different (*p* > 0.05). The FE model showed moderate heterogeneity between all studies comparing olanzapine vs. quetiapine and it was low in most subgroups (*I*^2^ = 0%), but it was substantial in subgroup 19.1.4 (*I*^2^ = 65%); the RE analysis was used for reporting on this outcome. Weight gain was significantly different (*p* < 0.001). Based on the reported RR, the type of APDs had a low impact on causing side effects in the study population (Table 9).

Gómez-Revuelta et al. [36] reported a higher rate of EPS with olanzapine (30%) than quetiapine (20%) (*p* > 0.05); however, Kahn et al. [37] found a lower prevalence with olanzapine (7%) than quetiapine (8%) (*p* > 0.05). The BAS scores in two studies [28,36] did not involve any significant differences (*p* > 0.05); however, exact scores were not reported. The heterogeneity of the included studies was moderate (*I*^2^ = 40%); the RE analysis was used for reporting on this outcome. There was no significant difference between the two drugs (*p* = 0.49); however, akathisia events were higher with quetiapine (16.7%) than olanzapine (14.3%). The reported RR (95% CI) was 0.81 (0.44, 1.48), indicating that the type of the drug had a low impact on causing treatment-emerged akathisia in the study population (Figure 8).

Heterogeneity between all studies comparing risperidone vs. quetiapine was low in all subgroups except in subgroups 20.1.2 (*I*^2^ = 54%) and 20.1.3 (*I*^2^ = 43%); the RE analysis was used for reporting on this outcome. Increased sleep duration was significantly different (*p* = 0.02). Most events were higher in risperidone except sleepiness, increased duration of sleep and constipation, which were higher in quetiapine; this may suggest that quetiapine showed better tolerability in the study sample. Based on the reported RR, the type of APDs had a low impact on causing side effects in the study population (Table 10).

At week 52, San et al. [25] reported slightly higher UKU psychiatric and neurological side effect scores for risperidone (2.5 ± 3.2 and 1.1 ± 1.8) than quetiapine (2.3 ± 1.4 and 0 ± 0). Both weight gain and glucose levels were higher for risperidone (+7 kg, 4.7 ± 0.3) than for quetiapine (+6 kg, 4.6 ± 0.7). However, all parameters were not significantly different (*p* > 0.05). The reported EPS percentage in Gómez-Revuelta et al.’s study [36] was higher for risperidone (40%) than quetiapine (20%), but the difference was not significant (*p*  =  0.043). The heterogeneity of the included studies was moderate (*I*^2^ = 40%); the RE analysis was used for reporting on this outcome. There was no significant difference (*p* = 0.49); however, akathisia events were higher with quetiapine (16.7%) than risperidone (14.3%). The reported RR (95% CI) was 0.81 (0.44, 1.48), indicating that the type of the drug had a low impact on causing treatment-emerged akathisia in the study population (Figure 9).

San et al. [25] concluded that the UKU psychiatric and neurological side effects were slightly higher in risperidone groups (2.5 ± 3.2 and 1.1 ± 1.8) than olanzapine groups (1.7 ± 1.8 and 0.3 ± 0.8). The results of weight gain were slightly greater in olanzapine groups (+9 kg, 4.6 ± 0.5) than risperidone groups (+7 kg, 4.7 ± 0.3) at week 52; however, there were no significant differences in any of the three outcomes (*p* > 0.05). The FE model showed low heterogeneity between all studies comparing olanzapine vs. risperidone, but it was moderate in subgroup 17.1.4 (*I*^2^ = 59%); the RE analysis was used for reporting on this outcome, where weight gain was found to have a significant difference (*p* = 0.03). Based on the reported RR, the type of APDs had a low impact on causing side effects in the study population (Table 11).

Only an abnormal cholesterol level, abnormal prolactin level, abnormal serum aspartate aminotransferase (AST) level and abnormal serum alanine aminotransferase (ALT) level were statistically significantly different (*p* < 0.05) between olanzapine and haloperidol [25,33,35,36,39]. The heterogeneity of the included studies was low (*I*^2^ = 0%). There was a significant difference (*p* < 0.001) with weight gain events, which were higher with olanzapine (63.4%) than haloperidol (36.6%). The reported RR (95% CI) was 1.59 (1.33, 1.91), indicating that the type of the drug had a high impact on causing weight gain in the study population (Figure 10).

The olanzapine group had less EPS events compared to haloperidol [33,38] (*p* < 0.001) and the severity of akathisia (BAS total score) was not significantly different (*p* > 0.05) [36], but actual scores were not reported. Heterogeneity of the included studies was low (*I*^2^ = 0%). The difference was significant (*p* < 0.001), but akathisia events were higher with olanzapine (26.7%) than haloperidol (8.6%). The reported RR (95% CI) was 3.19 (1.70, 6.00), indicating that the type of the drug had a high impact on causing akathisia in the study population (Figure 11).

The prevalence of side effects for the two APDs was not significantly different [25,36] in all measured outcomes (*p* > 0.05) between haloperidol and quetiapine. Figure 12 shows that the heterogeneity of the included studies was low (*I*^2^ = 0%) and that the difference was significant (*p* = 0.02), with the akathisia events being higher with haloperidol (26.7%) than quetiapine (14.6%). The reported RR (95% CI) was 1.81 (1.09, 3.01), indicating that the type of the drug had a high impact on causing treatment-emerged akathisia in the study population.

Only three studies [25,36,38] compared haloperidol and ziprasidone, and reported that there was no significant difference (*p* > 0.05) in all measured outcomes. Figure 13 shows that the heterogeneity of the included studies was low (*I*^2^ = 0%) and that the difference was significant between the two drugs (*p* = 0.03), with weight gain events being higher with haloperidol (50%) than ziprasidone (34.4%). The reported RR (95% CI) was 1.45 (1.03, 2.04), indicating that the type of the drug had a high impact on causing weight gain in the study population.

Three studies [25,36,37] reported no statistically significant differences between olanzapine and ziprasidone (*p* = 0.05) in all measured outcomes. Figure 14 shows that the heterogeneity of the included studies was low (*I*^2^ = 0%) and that the difference was significant (*p* < 0.001) in weight gain with higher events with olanzapine (81%) compared to ziprasidone (34.4%). The reported RR (95% CI) was 2.26 (1.69, 3.03), indicating that the type of the drug had a high impact on causing weight gain in the study population.

Additionally, one study [36] reported that aripiprazole administration was more likely to be associated with akinesia (*p*  =  0.004)). In two studies [27,38], the authors found that the clozapine group experienced fewer side effects compared to the chlorpromazine group at 52 weeks but it was not significant (*p* = 0.05). Gómez-Revuelta et al. [36] reported EPS was higher with risperidone (40%) than aripiprazole (23.8%) but it was not significant (*p*  =  0.456). Kahn et al. [37] found a higher prevalence of EPS with haloperidol than any SGA groups (34% vs. 6–17%; *p* < 0.001), and higher proportions of patients on haloperidol or ziprasidone experienced akathisia than with other APDs (26–28% vs. 10–16%; *p* < 0.01).

#### 7.2.2. Number Needed to Treat to Cause Harm (NNH) Calculation

The NNH calculation was based on the principle of direct comparison between two drugs rather than a drug vs. placebo. Aripiprazole caused less diminished sexual desire events (ARR = 0.0782, *p* = 0.01) than risperidone after short-term treatment. It also caused less sleepiness events (ARR = 0.0988, *p* = 0.03), less increased duration of sleep events (ARR = 0.13, *p* = 0.003), less rigidity events (ARR = 0.09, *p* = 0.02), less erectile dysfunction (ARR = 0.07, *p* = 0.005) and less ejaculatory dysfunction (ARR = 0.06, *p* = 0.02) than ziprasidone after long-term treatment. Additionally, aripiprazole caused less sleepiness events (ARR = 0.172, *p* < 0.001), less increased duration of sleep events (ARR = 0.016, *p* = 0.001), less tremor events (ARR = 0.092, *p* = 0.04) and less akathisia events (ARR = 0.092, *p* = 0.05) than quetiapine after long-term treatment.

Quetiapine caused less galactorrhoea events (ARR = 0.043, *p* = 0.03) and less erectile dysfunction events (ARR = 0.086, *p* = 0.002) than aripiprazole. It also caused less rigidity events (ARR = 0.093, *p* = 0.03), less vertigo events (ARR = 0.0769, *p* = 0.05) and less amenorrhoea events (ARR = 0.106, *p* = 0.006) than ziprasidone. Additionally, it caused less weight gain events (ARR = 0.185, *p* = 0.001) than olanzapine and less increased duration of sleep events (ARR = 0.013, *p* = 0.02) than risperidone after long-term treatment.

Ziprasidone caused less sleepiness events (ARR = 0.066, *p* = 0.07), less weight gain events (ARR = 0.174, *p* = 0.003) and one less akathisia event (ARR = 0.533, *p* = 0.005) than quetiapine. It also caused less weight gain events (ARR = 0.156, *p* = 0.03) than haloperidol and olanzapine (ARR = 0.467, *p* < 0.001).

Risperidone caused less weight gain (ARR = 0.108, *p* = 0.03) than olanzapine. Haloperidol caused less weight gain events (ARR = 0.268, *p* < 0.001) and akathisia events (ARR = 0.181, *p* = 0.0003) than olanzapine and less akathisia events than quetiapine (ARR = 0.114, *p* = 0.02) after long-term use of APDs.

### 7.3. Discontinuation of APDs

San et al. [25] showed olanzapine had a longer period of use (260.2 days) than quetiapine (187.1), which was similar in Gómez-Revuelta et al.’s study [36] (855 days vs. 60 days) (*p* < 0.05). The FE model showed low heterogeneity between studies comparing quetiapine vs. olanzapine and it was low for most of the subgroups’ analyses (*I*^2^ = 0–22%), but it was considerable in subgroup 56.1.3 (*I*^2^ = 84%); the RE analysis was used for reporting on this outcome. One subgroup had statistically significant differences between the two APDs (lack of efficacy, *p* < 0.001). Based on the reported RR, the type of APDs had a low impact on causing treatment discontinuation in the study population, which might be due to the patients’ factors (Table 12).

Time to discontinuation was longer with risperidone than with quetiapine (*p* < 0.05) (786 days vs. 60 days [36] and 206.2 days vs. 187.1 days [25]). Heterogeneity between studies comparing quetiapine vs. risperidone was low in all subgroups (*I*^2^ = 0%) and considerable in subgroup 55.1.1 (*I*^2^ = 86%); the RE analysis was used for reporting on this outcome. Discontinuation of APDs due to side effects was statistically significantly different between the two APDs (*p* = 0.02); the RR showed that the type of APDs had a low impact on causing treatment discontinuation in the study population (Table 13).

San et al. [25] reported that time to discontinuation for ziprasidone was shorter than quetiapine (142.7 vs. 187.1), which was not significantly different between the two APDs (*p* > 0.05). There was heterogeneity between all studies in all subgroups (*I*^2^ = 0–2%). Discontinuation due to lack of efficacy and side effects involved significant differences (*p* < 0.001). Based on the reported RR, the type of APDs had a high impact on causing treatment discontinuation in the study population (Table 14).

In one study, haloperidol had a shorter time to discontinuation (125 days) compared to quetiapine (187.1 days) [25], but in another study [35], it was shorter for quetiapine (60 days) than haloperidol (295 days) and it was significant (*p* < 0.05). Heterogeneity between studies comparing quetiapine and haloperidol in four subgroups was low (*I*^2^ = 0%) but considerable in subgroup 58.1.1 (*I*^2^ = 82%); the RE analysis was used for reporting on this outcome. Discontinuation due to side effects involved significant differences between the two APDs (*p* = 0.01). Based on the reported RR, the type of APDs had a high impact on causing treatment discontinuation in the study population (Table 15).

The mean time to any cause of discontinuation in the quetiapine-treated group [36] was only 60 days vs. 452 days for aripiprazole (*p* < 0.05); however, the gap between the two drugs was smaller in another study [39] (77.24 days vs. 106.71 days, respectively, *p* < 0.05). Heterogeneity between all studies comparing quetiapine and aripiprazole was low in all subgroups (*I*^2^ = 0–9%). Discontinuation due to lack of efficacy (*p* < 0.001) and lack of compliance (*p* = 0.04) was significantly different between the two APDs. Based on the reported RR, the type of APDs had a high impact on causing treatment discontinuation in the study population (Table 16).

In one study [36], a longer time (in days) to discontinuation was reported in the aripiprazole group (452 days) than in the ziprasidone group (251 days), but it was slightly shorter in aripiprazole-treated groups (106.71) than in the ziprasidone group (129.88) in another study [39], which was not significantly different (*p* > 0.05). There was heterogeneity between all studies comparing aripiprazole and ziprasidone in all subgroups (*I*^2^ = 0%). Discontinuation due to lack of compliance and side effects (*p* < 0.001) was statistically significantly different between the two APDs. Based on the reported RR, the type of APDs had a high impact on causing treatment discontinuation in the study population (Table 17).

The mean discontinuation time (in days) seen in the olanzapine group (855 days) and risperidone group (786 days) was not significantly different (*p* > 0.05) in one study [35], but it was significant (*p* < 0.05) in another study [25] where risperidone-treated patients had a lower (206) number of days than in the olanzapine group (260). Heterogeneity was moderate between all studies comparing risperidone and olanzapine in subgroup 61.1.2 (*I*^2^ = 41%); the RE analysis was used for reporting on this outcome with no significant differences between the two drugs in lack of efficacy (*p* = 0.65) and side effect (*p* = 0.25) subgroups. Based on the reported RR, the type of APDs had a high impact on causing treatment discontinuation in the study population (Table 18).

The time to discontinuation was reported in two studies [25,36] for risperidone and ziprasidone (786 days vs. 251 days and 206 vs. 142.7, respectively), which was significantly different (*p* < 0.05). Heterogeneity between all studies comparing risperidone and ziprasidone in all subgroups was moderate (*I*^2^ = 54, 44, 35%); the RE analysis was used for reporting on this outcome. Only lack of efficacy (*p* = 0.90), side effects (*p* = 0.42) and drop out (*p* = 0.59) were significantly different between the two drugs. Based on the reported RR, the type of APDs had a high impact on causing treatment discontinuation in the study population (Table 19).

Patients on haloperidol (295 days) had less time (in days) to discontinuation [36] compared to risperidone (786 days), which was similar to one other study [25] (125 days vs. 206 days), which was significant in both studies (*p* < 0.05). Heterogeneity between all studies comparing risperidone and haloperidol for all subgroups (*I*^2^ = 0%) was low. Only lack of efficacy (*p* = 0.30), side effects (*p* = 0.81) and drop out (*p* = 0.46) were significantly different. Based on the reported RR, the type of APDs had a high impact on causing treatment discontinuation in the study population (Table 20).

Only San et al. [25] showed the mean time to discontinuation in the ziprasidone group was less (142.7 days) than in the olanzapine group (260.2 days), which was significantly different between the two APDs (*p* < 0.05). Heterogeneity between all studies comparing olanzapine and ziprasidone was substantial in the 74.1.4 subgroup (*I*^2^ = 70%); the RE analysis was used for reporting on this outcome. Discontinuation due to lack of efficacy (*p* = 0.01), side effects (*p* < 0.001) and lack of compliance (*p* = 0.05) was significantly different. Based on the reported RR, the type of APDs had a high impact on causing treatment discontinuation in the study population (Table 21).

San et al. [25] concluded that the olanzapine group mean time to discontinuation was longer (260 days) than with haloperidol (125 days) (*p* > 0.05). Heterogeneity between all studies comparing olanzapine and haloperidol was considerable in subgroup 75.1.4 (*I*^2^ = 68%); the RE analysis was used for reporting on this outcome. Lack of efficacy (*p* < 0.001) and side effects (*p* = 0.001) were significantly different. Based on the reported RR, the type of APDs had a high impact on causing treatment discontinuation in the study population (Table 22).

The mean time to discontinuation was longer for ziprasidone (142.7 days) than haloperidol (125 days) in one study [26] but it was not in another [35] (ziprasidone: 251 days vs. haloperidol: 295 days). There was heterogeneity between all studies comparing haloperidol and ziprasidone in all subgroups (*I*^2^ = 0–33%). Discontinuation due to lack of compliance (*p* = 0.01) was the only significantly different reason. Based on the reported RR, the type of APDs had a high impact on causing treatment discontinuation in the study population (Table 23).

Girgis et al. [38] reported that 26% of the clozapine group and 10% of the chlorpromazine group remained on their assigned treatments for the duration of the study (*p* = 0.01). The median amount of time until first discontinuation was longer in the clozapine group (39 months) over the chlorpromazine group (23 months); the difference was significantly longer (*p* = 0.01). Reasons for discontinuation in the chlorpromazine group were withdrawn consent: 7.5%, side effects: 1.25%, lack of efficacy: 1.25%, death: 2.5%, did not attend follow up: 10% and imprisonment: 1.25%. Reasons for discontinuation in the clozapine group were withdrawn consent: 7.5%, side effects: 1.25%, death: 2.5%, lost to follow up: 8.75% and escape from the hospital: 1.25%.

#### Number Needed to Treat (NNT) to Cause Discontinuation of APDs

Olanzapine caused less total discontinuation events (ARR = 0.087, *p* = 0.030) than quetiapine. Olanzapine caused less total discontinuation events (ARR = 0.146, *p* = 0.02) than ziprasidone and haloperidol (ARR = 0.203, *p* = 0.002). Olanzapine caused less discontinuation due to lack of efficacy (ARR = 0.178, *p* < 0.001) than quetiapine. Olanzapine caused less discontinuation due to lack of efficacy (ARR = 0.0977, *p* = 0.01) than ziprasidone, and caused less discontinuation due to lack of efficacy (ARR = 0.117, *p* < 0.001) and side effects (ARR = 0.084, *p* = 0.001) than haloperidol.

Quetiapine caused less discontinuation due to side effects than risperidone (ARR = 0.133, *p* = 0.02), ziprasidone (ARR = 0.177, *p* < 0.001) and haloperidol (ARR = 0.063, *p* = 0.01), and caused less discontinuation due to lack of compliance (ARR = 0.086, *p* = 0.04) than aripiprazole.

Ziprasidone caused less total discontinuation events (ARR = 0.11, *p* = 0.0003) than quetiapine and haloperidol (ARR = 0.063, *p* = 0.05). Ziprasidone caused less discontinuation due to lack of efficacy (ARR = 0.345, *p* < 0.001) than quetiapine, caused less discontinuation due to lack of compliance (ARR = 0.145, *p* = 0.0005) than aripiprazole and caused less discontinuation due to lack of compliance (ARR = 0.098, *p* = 0.01) than haloperidol.

Aripiprazole caused less total discontinuation events (ARR = 0.141, *p* = 0.02) than ziprasidone, and caused less total discontinuation events (ARR = 0.375, *p* = 0.004) than quetiapine. Aripiprazole caused less discontinuation due to lack of efficacy (ARR = 0.345, *p* < 0.001) than quetiapine, caused less discontinuation due to side effects (ARR = 0. 145, *p* < 0.001) than ziprasidone and caused less discontinuation due to side effects (ARR = 0.1869, *p* < 0.001) and lack of compliance (ARR = 0.071, *p* = 0.05) than olanzapine.

### 7.4. Risk of Bias

Included were 21 randomised controlled studies to compare the APD outcomes with one another. Among the total of 21 studies, 11 studies remained at a low risk of bias. In addition, Stauffer et al. [42] showed participants were predominantly male; however, it was unclear whether this had affected the outcome of the study, which was considered under ‘other’ bias. Perkins et al.’s study [43] could not be included in the narrative or statistical analysis as it did not report data for the different APDs but rather the class in general. In addition, it was not possible to statistically analyse Lieberman et al. [32] and Girgis et al.’s [38] data for clozapine and chlorpromazine due to the lack of population size (Figure 15).

## 8. Review Limitations

This study has some limitations, which were mitigated to the minimum impact possible on the reported results. There was disparity in the way numerical data were reported; as such, in some occasions, a statistical analysis could not be performed. The number of measured outcomes and the scales they were measured on also varied between studies. The systematic review was time-limited and not a live review, as it is part of a degree not for a provider; as such, only studies up to 2022 were included in the analysis; updated systematic reviews are planned for the near future. In some subgroups, only two studies were available, which may have impacted the generalisation of the finding to a wider population. This systematic review did not consider confounders such as age, gender or ethnicity, which are known to possibly impact the severity of side effects and response to treatment and should be considered in future systematic reviews. Moreover, this study only included patients who were either APD-naïve or with short-term history of APD use (0–16 weeks); therefore, a future systematic review may consider investigating discontinuation reasons in patients with chronic psychosis.

## 9. Discussion

This systematic review gives a comprehensive view about the efficacy and tolerability of APDs in direct paired comparison. Twenty-one RCTs were included, and APDs were paired and compared. The fixed model analysis was used when the heterogeneity in all subgroups was <30%, and the random model analysis was used when the heterogeneity was >30%. A minimum of two studies were required to conduct a statistical analysis. A statistical analysis was not possible for all short-term uses of APDs (0–12 weeks) based on the data available from the selected studies; accordingly, data from the included studies were narratively reported in Section 2 of this paper. Only three pairs of APDs made it possible to statistically compare using the mean difference to measure symptoms’ improvement on one or more assessment tools (PANSS, CGI, BPRS, SANS, SAPS, CDSS and YMRS) after long-term use (>12 weeks. APDs were paired and compared based on their side effects; the use of medications from other classes such as hypnotics, anticholinergics or any others to manage those side effects, which allow the continuation of therapy (data reported in Section 3, part 3) and discontinuation rate and reasons (data reported in Section 3, part 4). A statistical analysis was not possible for some measured outcomes (side effects or concomitant medication used to manage them, discontinuation rate or reasons) due to the data available from the selected studies. Eight systematic reviews were identified to compare this systematic review’s findings with their findings. Eight drugs were the most compared in those systematic reviews and were compared to the findings from this systematic review. Any data from this systematic review will be stated as a numerical summary or final outcome (*p*-value or total score) and labelled as ‘this review’ to avoid repeating the result sections stated previously in the results. Additionally, where comparison with the literature was made for one drug, e.g., olanzapine vs. risperidone, it was not repeated again under risperidone to prevent repetition. See Appendix A.

### 9.1. Olanzapine

Huhn et al. [45] showed the overall reduction in symptoms was significantly more for olanzapine (−0.56 [−0.62 to −0.50]) than the placebo and compared with haloperidol (−0.47 [−0.53 to −0.41]), quetiapine (−0.42 [−0.50 to −0.33]), aripiprazole (−0.41 [−0.50 to −0.33]) and ziprasidone (−0.41 [−0.50 to −0.32]). Negative symptoms were significantly lower for olanzapine (−0.45 [−0.51 to −0.39]) than ziprasidone (−0.33 [−0.43 to −0.23]) and aripiprazole (−0.33 [−0.41 to −0.20]), quetiapine (−0.31 [−0.38 to −0.24]) and haloperidol (−0.29 [−0.35 to −0.23]). Depressive symptoms were significantly lower with olanzapine (−0.37 [−0.46 to −0.29]) compared with other APDs. Hartling et al. [46] showed a clinically important benefit of olanzapine over haloperidol for negative PANSS (*p* < 0.001), SANS symptoms (*p* = 0.002) and MADRS (*p* = 0.001). The PANSS total and also prevalence of symptoms’ improvement were higher with olanzapine than haloperidol (*p* = 0.02); conversely, haloperidol had a clinically important benefit over olanzapine on SAPS (*p* < 0.001). The results from Davis et al. [47] concluded that CGI, PANSS and BPRS did not show a difference in efficacy between olanzapine, amisulpride and risperidone, but it was statistically significant between them and FGAs (*p* < 0.001), proving their higher efficacy. The overall symptom changes in Leucht et al.’s study [48] showed significant favourability of olanzapine with the comparative antipsychotics: olanzapine vs. haloperidol, −0.14 (−0.21 to −0.08); olanzapine vs. quetiapine, −0.15 (−0.25 to −0.06); olanzapine vs. aripiprazole, −0.16 (−0.25 to −0.07); and olanzapine vs. ziprasidone, −0.20 (−0.29 to −0.10). Zhu et al. [49] showed that the overall reduction in symptoms showed to be significantly more for olanzapine (−0.25 [−0.39 to −0.12]) compared with haloperidol; reduction in negative symptoms showed as significantly higher for olanzapine over haloperidol (0.31 [0.13 to 0.48]) and risperidone (0.20 [0.03 to 0.37]).

Huhn et al. [45] reported that olanzapine (2.78 [2.44 to 3.13]) produced significantly more weight gain against the placebo and compared with many other drugs such as quetiapine (1.94 [1.42 to 2.45]), clozapine (1.89 [0.36 to 3.43]), risperidone (1.44 [1.05 to 1.83]), amisulpride (0.84 [0.14 to 1.53]), haloperidol (0.54 [0.15 to 0.95]), aripiprazole and ziprasidone (−0.16 [−0.73 to 0.40]). Olanzapine (4.29 [1.91 to 6.68]) caused significantly more QTc prolongation than the placebo, and it involved significantly more sedating than the placebo (2·17 [1.93 to 2.40]) and significantly higher anticholinergic side effects than the placebo (1.94 [1.46 to 2.48]). In Leucht et al.’s study [48], olanzapine (0.74 [0.67 to 0.81]) produced significantly more weight gain than all other drugs. Olanzapine (1.00 [0.73 to 1.33]) did not cause significantly more EPS than the placebo. Kishimoto et al. [50] reported sedation and/or somnolence with olanzapine was significantly less than clozapine (*p* < 0.001) and quetiapine, and olanzapine was significantly associated more with sedation than risperidone (*p* = 0.010). Parkinsonism symptoms showed to be significantly associated more with risperidone than olanzapine (*p* < 0.001). Zhang et al. [51] concluded that both olanzapine and risperidone increased weight significantly more than haloperidol (*p* < 0.01) in the short-term treatment analysis. It was also shown in both short-term treatment (*p* < 0.05) and long-term treatment (*p* < 0.001) that akathisia was less likely with olanzapine and risperidone than haloperidol. This study also reported that EPS was not as frequent with both olanzapine (*p* < 0.001) and risperidone (*p* < 0.001) compared with haloperidol. The authors reported that in the short-term analysis, patients on haloperidol required less anticholinergics (*p* < 0.001), benzodiazepines (*p* = 0.02) and beta-blockers (*p* < 0.01) compared to olanzapine. Zhu et al. [49] found that olanzapine was associated with a lower rate of parkinsonian symptoms than haloperidol (0.10 [0.03 to 0.29]) and risperidone (0.24 [0.07 to 0.78]).

Katona et al. [52] found that olanzapine had a lesser discontinuation rate compared to amisulpride (RR = 0.69), aripiprazole (RR = 0.88), haloperidol (RR = 0.58), quetiapine (RR = 0.72) and risperidone (RR = 0.71). Additionally, Kishimoto et al. [50] showed that olanzapine had a significantly lower all-cause of continuation as compared with quetiapine (*p* < 0.001), risperidone (*p* < 0.001), aripiprazole (*p* = 0.006) and ziprasidone (*p* < 0.001) after long-term use. Zhang et al. [51] reported that discontinuation due to inefficacy (*p* = 0.04) and discontinuation due to intolerability (*p* < 0.001) were higher with haloperidol than olanzapine. In addition, Zhang et al. [51] reported that after short-term use, olanzapine (*p* = 0.001) had a lesser discontinuation rate than risperidone (*p* = 0.02) and quetiapine (*p* < 0.01). Huhn et al. [45] reported that olanzapine, 0.69 (0.65 to 0.74), had lower all-cause discontinuation rates than the placebo and haloperidol, 0.53 (0.40 to 0.70); quetiapine, 0.70 (0.51 to 0.95); aripiprazole, 0.76 (0.64 to 0.90); and ziprasidone, 0.65 (0.53 to 0.79). In Zhu et al.’s study [49], haloperidol was associated with a significantly higher discontinuation rate compared with olanzapine (1.83 [1.23 to 2.74]).

Summary of findings: Based on the above findings, while limited to the population of the included studies, olanzapine improved symptoms in more patients (second to clozapine), was better tolerated except for weight gain and allowed more patient treatment continuation and for a longer period than other FGAs and other APDs except for clozapine, which is comparable to that reported in the reviewed systematic reviews.

### 9.2. Quetiapine

Huhn et al. [45] reported that quetiapine (−0.42 [−0.50 to −0.33]) had a lower overall reduction in symptoms than clozapine (−89 [−1.08 to −0.71]), amisulpride (−0.73 [−0.89 to −0.58]), olanzapine (−0.56 [−0.62 to −0.50]) and risperidone (−0.55 [−0.62 to −0.48]). Also, negative symptoms were higher for quetiapine (−0.31 [−0.38 to −0.24]) than for clozapine (−0.62 [−0.84 to −0.39]), amisulpride (−0.50 [−0.64 to −0.37]), olanzapine (−0.45 [−0.51 to −0.39]) and, to a lesser extent, risperidone (−0.37 [−0.43 to −0.31]).

Huhn et al. [45] showed that quetiapine (−1.17 [−4.52 to 2.27]) did not show any significant differences compared to the placebo in EPS symptoms. They also reported that quetiapine (3.43 [0.94 to 6.00]) caused significantly more QTc prolongation than the placebo and it was significantly more sedating and one of the most sedating (3.27 [2.61 to 4.22]) compared to the placebo. Quetiapine (3.89 [2.83 to 5.56]) was reported to have significantly higher anticholinergic side effects than the placebo. Leucht et al. [48] reported that quetiapine (0.43 [0.34 to 0.53]) produced significantly more weight gain than haloperidol (0.09 [−0.00 to 0.17]), ziprasidone (0.10 [−0.02 to 0.22]), aripiprazole (0.17 [0.05 to 0.28]) and amisulpride (0.20 [0.05 to 0.35]). Quetiapine (1.01 [0.68 to 1.44]) did not cause significantly more EPS than the placebo. In addition, quetiapine was significantly less associated with QT prolongation compared with ziprasidone (−0.24 [−0.38 to −0.10]). Kishimoto et al. [50] showed a higher prolactin level increase with quetiapine (*p* = 0.006) than amisulpride. Moreover, dyskinesia was significantly associated with quetiapine compared to ziprasidone (*p* = 0.030).

Kishimoto et al. [50] reported that quetiapine involved significantly higher all-cause discontinuation compared with ziprasidone (*p* = 0.031). Zhang et al. [51] reported discontinuation due to patient decision and non-adherence, and only quetiapine showed a significantly lower rate than haloperidol (*p* < 0.05). Huhn et al. [45] reported that quetiapine, 0.85 (0.82 to 0.89), was higher than the placebo.

Summary of findings: Based on the above findings, while limited to the population of the included studies, quetiapine improved symptoms in more patients than those treated with FGAs but in less patients than those treated with clozapine, olanzapine, aripiprazole and ziprasidone. It was better tolerated except for weight gain and allowed more patient treatment continuation and for a longer period than other FGAs and other APDs except for clozapine, olanzapine, aripiprazole and ziprasidone, which is comparable to that reported in the reviewed systematic reviews.

### 9.3. Ziprasidone and Aripiprazole

Leucht et al. [48] reported aripiprazole did not cause significantly increased prolactin concentrations compared with the placebo. Aripiprazole was not associated with significant QTc prolongation compared with the placebo and was better than olanzapine, −0.21 (−0.37 to −0.05); ziprasidone, −0.40 (−0.55 to −0.26); risperidone, −0.25 (−0.40 to −0.10); and amisulpride, −0.65 (−0.93 to −0.35). In addition, ziprasidone involved a higher risk to cause QTc prolongation than risperidone, 0.16 (0.04 to 0.29), and haloperidol, 0.30 (0.21 to 0.40). Kishimoto et al. [50] reported aripiprazole, *p* < 0.001, caused less weight gain than olanzapine. Katona et al. [44] found that aripiprazole had less discontinuation compared to amisulpride (RR = 0.78) and risperidone (RR = 0.83). Similarly, Zhang et al. [51] found that the outcomes of short-term treatment with aripiprazole included all-cause discontinuation over ziprasidone. Huhn et al. [45] reported that aripiprazole, 0.80 (0.73 to 0.86), and ziprasidone, 0.88 (0.80 to 0.96), had higher all-cause discontinuation rates than the placebo.

Summary of findings: Based on the above findings, while limited to the population of the included studies, aripiprazole and ziprasidone were similar in improving symptoms in more patients than those treated with FGAs but to a lesser extent than that achieved with clozapine and olanzapine. They were better tolerated and allowed more patient treatment continuation and for a longer period than other FGAs and other APDs except for clozapine and olanzapine, which is comparable to that reported in the reviewed systematic reviews.

### 9.4. Clozapine

Huhn et al. [45] reported overall reduction in symptoms being significantly associated more with clozapine (−89 [−1.08 to −0.71]) than the placebo and compared with other antipsychotics including haloperidol (−0.47 [−0.53 to −0.41]), quetiapine (−0.42 [−0.50 to −0.33]), aripiprazole (−0.41 [−0.50 to −0.33]) and ziprasidone (−0.41 [−0.50 to −0.32]). Negative symptoms were significantly lower for clozapine (−0.62 [−0.84 to −0.39]) than many other drugs including ziprasidone (−0.33 [−0.43 to −0.23]), aripiprazole (−0.33 [−0.41 to −0.20]), quetiapine (−0.31 [−0.38 to −0.24]) and haloperidol (−0.29 [−0.35 to −0.23]). Depressive symptoms were significantly lower for clozapine (−0.52 [−0.82 to −0.23]) compared with all other APDs. The overall symptom changes in Leucht et al.’s [48] study showed clozapine was significantly associated more with the improvement in overall symptom change: clozapine vs. olanzapine (−0.29 [−0.44 to −0.14]), clozapine vs. risperidone (−0.32 [−0.47 to −0.16]), clozapine vs. haloperidol (−0.43 [−0.58 to −0.28]), clozapine vs. quetiapine (−0.44 [−0.61 to −0.28]), clozapine vs. aripiprazole (−0.45 [−0.62 to −0.28]) and clozapine vs. ziprasidone (−0.49 [−0.66 to −0.31]). In the systematic review by Lieberman et al. [33], the authors found that only the SANS total score was statistically significantly different between clozapine and chlorpromazine after short-term use (*p* = 0.01), which is similar to Hartling et al.’s [47] systematic review where clozapine and chlorpromazine were statistically significantly different on the BPRS scale (*p* = 0.001).

Leucht et al. [48] reported clozapine (0.65 [0.31 to 0.99]) and chlorpromazine (0.55 [0.34 to 0.76]) produced significantly more weight gain than the placebo and quetiapine (0.43 [0.34 to 0.53]) and risperidone (0.42 [0.33 to 0.50]), haloperidol (0.09 [−0.00 to 0.17]), ziprasidone (0.10 [−0.02 to 0.22]), aripiprazole (0.17 [0.05 to 0.28]) and amisulpride (0.20 [0.05 to 0.35]). Leucht et al. [49] showed that clozapine (0.3 [0.12 to 0.62]), olanzapine (1.00 [0.73 to 1.33]), quetiapine (1.01 [0.68 to 1.44]), aripiprazole (1.20 [0.73 to 1.85]) and amisulpride (1.60 [0.88 to 2.65]) did not cause significantly more EPS than the placebo. Kishimoto et al. [50] showed the prolactin increase with clozapine (*p* < 0.001), olanzapine (*p* < 0.001), quetiapine (*p* < 0.001) and ziprasidone (*p* < 0.001) was significantly less than with risperidone. Additionally, Davis et al. [47] found that there was a statistically significant difference among clozapine compared to all FGAs (*p* < 0.001). Huhn et al. [46] reported clozapine (1.89 [0.36 to 3.43) involved more weight gain than the placebo. Clozapine showed it was significantly more sedating than the placebo. Clozapine (2.21 [1.26 to 3.47]) had significantly higher anticholinergic side effects than the placebo. Huhn et al. [45] reported that anti-Parkinson’s medication was used with clozapine (0.46 [0.19 to 0.88]) compared to the placebo. They also reported on all-cause discontinuation rates compared to the placebo (from the most favourable to the least): clozapine (0.76 [0.59 to 0.92]), risperidone (0.82 [0.80 to 0.85]), aripiprazole (0.80 [0.73 to 0.86]), quetiapine (0.85 [0.82 to 0.89]), ziprasidone (0.88 [0.80 to 0.96]) and haloperidol (0.90 [0.85 to 0.95]). Girgis et al. [38] reported that reasons for discontinuation in the chlorpromazine group were side effects (1.25%), lack of efficacy (1.25%) and death (2.5%) compared to side effects (1.25%) and death (2.5%) for clozapine.

Summary of findings: Based on the above findings, while limited to the population of the included studies, clozapine improved symptoms in more patients, was better tolerated except for weight gain and allowed more patient treatment continuation and for a longer period than other APDs, which is comparable to that reported in the reviewed systematic reviews.

### 9.5. Amisulpride

Huhn et al. [45] showed the overall reduction in symptoms was significantly more for amisulpride (−0.73 [−0.89 to −0.58]), clozapine (−89 [−1.08 to −0.71]), olanzapine (−0.56 [−0.62 to −0.50]) and risperidone (−0.55 [−0.62 to −0.48]) compared with the placebo and haloperidol (−0.47 [−0.53 to −0.41]), quetiapine (−0.42 [−0.50 to −0.33]), aripiprazole (−0.41 [−0.50 to −0.33]) and ziprasidone (−0.41 [−0.50 to −0.32]). Also, negative symptoms were significantly lower for amisulpride (−0.50 [−0.64 to −0.37]), clozapine (−0.62 [−0.84 to −0.39]), olanzapine (−0.45 [−0.51 to −0.39]) and, to a lesser extent, risperidone (−0.37 [−0.43 to −0.31]) than ziprasidone (−0.33 [−0.43 to −0.23]) and aripiprazole (−0.33 [−0.41 to −0.20]), quetiapine (−0.31 [−0.38 to −0.24]) and haloperidol (−0.29 [−0.35 to −0.23]). Depressive symptoms were significantly lower for amisulpride (−0.44 [−0.60 to −0.28]), olanzapine (−0.37 [−0.46 to −0.29]) and clozapine (−0.52 [−0.82 to −0.23]) compared with other APDs. Leucht et al.’s [48] study shows favourability of amisulpride with the comparative antipsychotics regarding symptoms’ improvement: amisulpride vs. haloperidol (−0.21 [−0.32 to −0.09]), amisulpride vs. aripiprazole (−0.23 [−0.37 to −0.08]) and amisulpride vs. ziprasidone (−0.26 [−0.41 to −0.12]).

Huhn et al. [45] showed amisulpride (14.10 [7.71 to 20.45]) caused significantly more QTc prolongation than the placebo. Amisulpride (1.56 [0.91 to 2.23]) and aripiprazole (1.46 [1.11 to 1.83]) were higher than the placebo but less sedating than other APDs. Leucht et al. [48] reported that amisulpride (1.60 [0.88 to 2.65]) did not cause significantly more EPS than the placebo. Kahn et al. [37] also found that the total discontinuation rate in their amisulpride group was similar to that in ziprasidone (37%) and olanzapine (28%) groups, but lower than haloperidol (61%) and quetiapine (48%) groups. In Huhn et al.’s study [45], amisulpride involved significantly lower all-cause discontinuation than haloperidol (0.53 [0.40 to 0.70]), quetiapine (0.70 [0.51 to 0.95]), aripiprazole (0.71 [0.51 to 0.96]) and ziprasidone (0.60 [0.43 to 0.83]).

Summary of findings: The finding from this study and the other included systematic reviews did not allow for sufficiently conducting a comprehensive comparison between amisulpride and clozapine, olanzapine, aripiprazole, ziprasidone, quetiapine, risperidone and haloperidol in side effects and discontinuation rate and reasons. However, based on the available data, amisulpride was effective in symptoms’ improvement and had less side effects than the compared FGAs and SGAs.

### 9.6. Risperidone

Zhu et al. [49] reported the overall reduction in symptoms was higher for risperidone (−0.14 [−0.27 to −0.01]) compared with haloperidol. Huhn et al. [45] found that the overall reduction in symptoms was significantly more for risperidone (−0.55 [−0.62 to −0.48]) than the placebo and compared with haloperidol (−0.47 [−0.53 to −0.41]), quetiapine (−0.42 [−0.50 to −0.33]), aripiprazole (−0.41 [−0.50 to −0.33]) and ziprasidone (−0.41 [−0.50 to −0.32]). Negative symptoms were significantly lower for risperidone (−0.37 [−0.43 to −0.31]) than ziprasidone (−0.33 [−0.43 to −0.23]) and aripiprazole (−0.33 [−0.41 to −0.20]), quetiapine (−0.31 [−0.38 to −0.24]) and haloperidol (−0.29 [−0.35 to −0.23]). Hartling et al. [46] found that total PANSS was significantly higher for risperidone than haloperidol (*p* < 0.001). Leucht et al. [48] concluded that risperidone was better when compared with other APDs: risperidone vs. haloperidol (−0.11 [−0.18 to −0.05]), risperidone vs. quetiapine (−0.13 [−0.22 to −0.03]), risperidone vs. aripiprazole (−0.13 [−0.23 to −0.03]) and risperidone vs. ziprasidone (−0.17 [−0.27 to 0.07]).

In Huhn et al.’s study [45], compared with the placebo, risperidone showed significantly more elevated prolactin levels (37.98 [34.64 to 41.38]); this difference was smaller but still significant with haloperidol (18.49 [15.60 to 21.39]) and olanzapine, 94.47 [1.60 to 7.38]). The use of anti-Parkinson’s medication was significantly worse regarding haloperidol than risperidone (1.80 [1.40 to 2.38]). Risperidone (4.77 [2.68 to 6.87]) caused significantly more QTc prolongation than the placebo. Also, risperidone (2.03 [1.67 to 2.51]) was significantly more sedating than the placebo. Risperidone (1.31 [1.03 to 1.72]) showed significantly higher anticholinergic side effects than the placebo. Leucht et al. [49] found that risperidone (0.42 [0.33 to 0.50]) produced significantly more weight gain than haloperidol (0.09 [−0.00 to 0.17]), ziprasidone (0.10 [−0.02 to 0.22]), aripiprazole (0.17 [0.05 to 0.28]) and amisulpride (0.20 [0.05 to 0.35]). Moreover, risperidone was among the least well tolerated drugs due to the significant differences in EPS compared with aripiprazole (1.83 [1.08 to 2.94]), quetiapine (0.49 [0.32 to 0.73]), clozapine (0.15 [0.06 to 0.30]) and olanzapine (0.48 [0.34 to 0.66]). Risperidone involved the highest prolactin level increase compared to haloperidol (−0.53 [−0.71 to −0.34]), ziprasidone (−0.98 [−1.24 to −0.72]), olanzapine (−1.09 [−1.28 to −0.90]), quetiapine (−1.28 [−1.50 to −1.06]) and aripiprazole (−1.45 [−1.71 to −1.18]). In addition, ziprasidone involved a higher risk for prolactin elevation than risperidone (0.16 [0.04 to 0.29]) and haloperidol (0.30 [0.21 to 0.40]). Kishimoto et al. [51] reported that parkinsonian side effects were significantly associated more with risperidone than olanzapine (*p* < 0.001). Zhang et al. [51] concluded that EPS was not as frequent with risperidone (*p* < 0.001) compared with haloperidol.

Kishimoto et al. [50] found that risperidone showed significantly lower all-cause discontinuation as compared with ziprasidone (*p* = 0.012). Leucht et al. [48] reported that risperidone involved significantly lower discontinuation compared with haloperidol (0.66 [0.58 to 0.76]) and ziprasidone (0.75 [0.61 to 0.91]).

Summary of findings: Based on the above findings, while limited to the population of the included studies, risperidone was more effective than all other FGAs in improving symptoms but less so than the effectiveness achieved with clozapine, olanzapine, aripiprazole, ziprasidone and quetiapine. It was better tolerated and allowed more patient treatment continuation and for a longer period than other FGAs but not SGAs, which is comparable to that reported in the reviewed systematic reviews.

### 9.7. Haloperidol

Hartling et al. [46] found that haloperidol had a clinically important benefit over olanzapine on positive symptom SAPS (*p* < 0.001). Huhn et al. [45] showed the use of anti-Parkinson’s medication was significantly worse regarding haloperidol than the placebo and clozapine (0.46 [0.19 to 0.88]), olanzapine (1.02 [0.79 to 1,30]), quetiapine (1.05 [0.78 to 1.48]), aripiprazole (1.32 [0.90 to 1.82]), amisulpride (1.46 [0.96 to 2.04]), ziprasidone (1.70 [1.23 to 2.46]) and risperidone (1.80 [1.40 to 2.38]). QTc prolongation was non-significant for haloperidol (1·69 [−0·23 to 3·64]) compared with the placebo. Haloperidol (1.50 [1.14 to 1.93]) had significantly higher anticholinergic side effects than the placebo. Leucht et al. [9] reported haloperidol (0.09 [−0.00 to 0.17]) had the least weight gain compared to the placebo and other APDs. Haloperidol (0.21 [0.14 to 0.31]) was associated with significantly more EPS than ziprasidone (0.34 [0.22 to 0.50]), aripiprazole (0.25 [0.15 to 0.39]), amisulpride (0.34 [0.19 to 0.54]), risperidone (0.44 [0.34 to 0.57]), clozapine (0.06 [0.02 to 0.13]) and olanzapine (0.21 [0.16 to 0.28]); this difference was non-significant with chlorpromazine (2.65 [1.33 to 4.76]). Also, haloperidol was associated with significantly more prolactin level increases than ziprasidone (−0.45 [−0.69 to −0.22]), olanzapine (−0.56 [−0.73 to −0.40]), quetiapine (−0.75 [−0.96 to −0.55]) and aripiprazole (−0.92 [−1.17 to −0.66]). Zhang et al. [51] concluded that the longer-term treatment analysis showed anticholinergic use remained significantly higher regarding any FGAs including haloperidol than any SGAs, while the short-term analysis showed that patients on haloperidol required less anticholinergics (*p* < 0.001), benzodiazepines (*p* = 0.02) and beta-blockers (*p* < 0.01) compared to olanzapine.

They also reported discontinuation due to inefficacy (*p* = 0.04) and intolerability (*p* < 0.001) was higher with haloperidol than olanzapine. Leucht et al. [48] reported that haloperidol was the worst compared to any other antipsychotics in all-cause discontinuation. The difference between haloperidol and quetiapine (1.32 [1.11 to 1.57]) and aripiprazole (1.33 [1.11 to 1.57]) was significant, favouring quetiapine and aripiprazole over haloperidol.

Summary of findings: Based on the above findings, while limited to the population of the included studies, haloperidol was more effective than all other FGAs, except for risperidone, in improving symptoms but less so than the effectiveness achieved with clozapine, olanzapine, aripiprazole and ziprasidone and quetiapine. It was better tolerated and allowed more patient treatment continuation and for a longer period than other FGAs, but was second to risperidone, but not SGAs, which is comparable to that reported in the reviewed systematic reviews.

## 10. Conclusions

This study concluded that in patients’ non-adherence to therapy, inadequate insight or knowledge of their condition or therapy and practitioners’ empathy are the most significant challenges encountered in the management of schizophrenia. Additionally, the nature of the medication used to treat schizophrenia, its safety profile, the disparity in individual treatment response and the extent of their experience with side effects result in patients requesting to stop treatment or practitioners deciding to cease therapy. This requires currently available therapy choices to be individualised to each patient, rather than applying a stepped-up or rigid guideline approach.

## Figures and Tables

**Figure 1 pharmacy-11-00175-f001:**
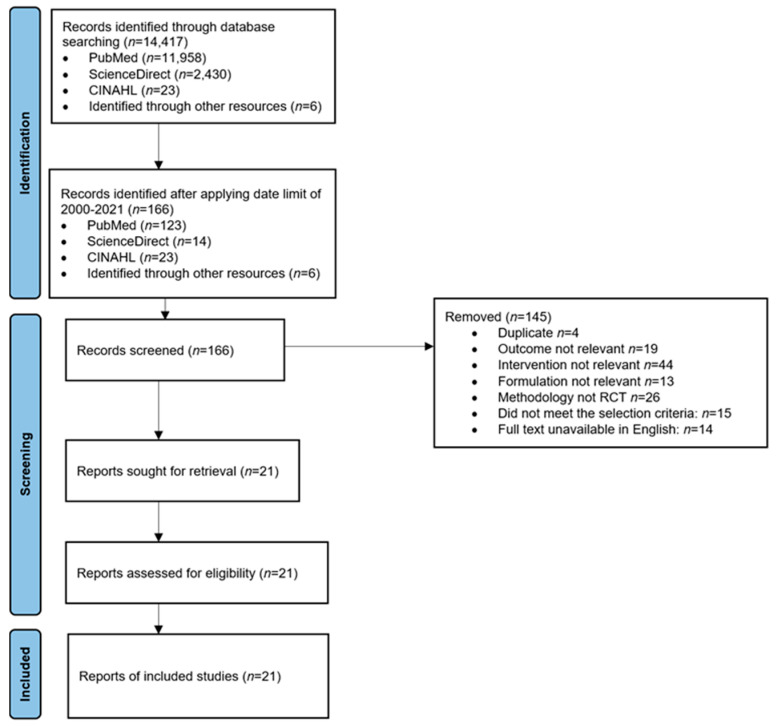
Study selection Prisma chart.

**Figure 2 pharmacy-11-00175-f002:**
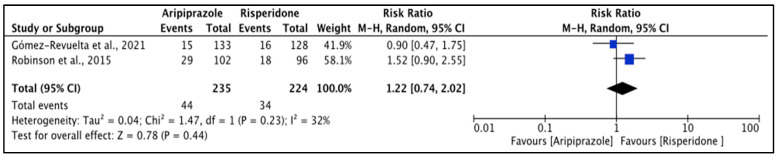
Short-term treatment of akathisia in aripiprazole vs. risperidone RE [30,31].

**Figure 3 pharmacy-11-00175-f003:**
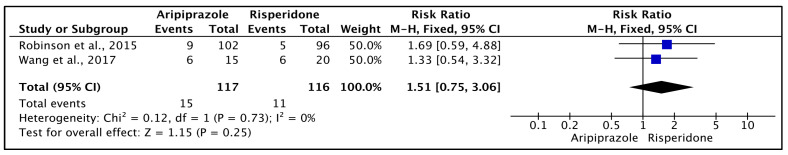
Treatment-emerged EPS in aripiprazole vs. risperidone FE [26,31].

**Figure 4 pharmacy-11-00175-f004:**
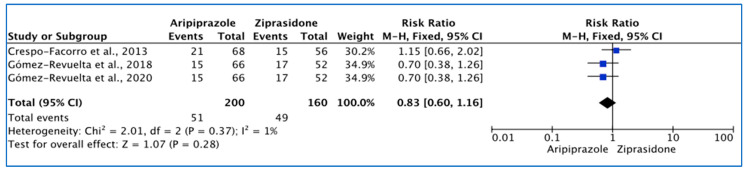
Long-term treatment-emergent akathisia in aripiprazole vs. ziprasidone, FE [34,35,36].

**Figure 5 pharmacy-11-00175-f005:**
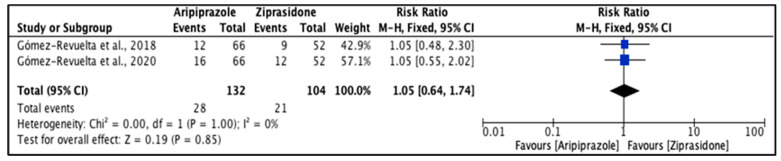
Long-term treatment-emergent EPS in aripiprazole vs. ziprasidone, FE [35,36].

**Figure 6 pharmacy-11-00175-f006:**
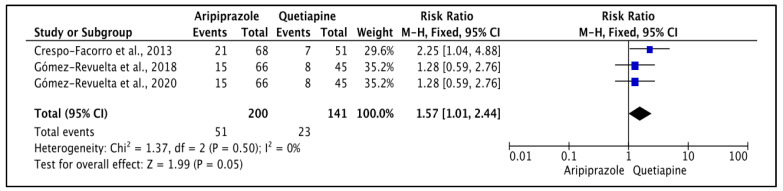
Treatment-emergent akathisia in aripiprazole vs. quetiapine in FE [35,36,39].

**Figure 7 pharmacy-11-00175-f007:**
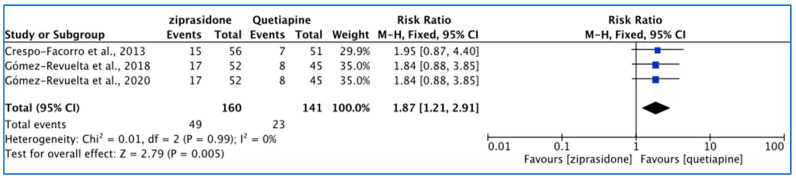
Long-term treatment-emergent akathisia in ziprasidone vs. quetiapine, FE [36,38,40].

**Figure 8 pharmacy-11-00175-f008:**
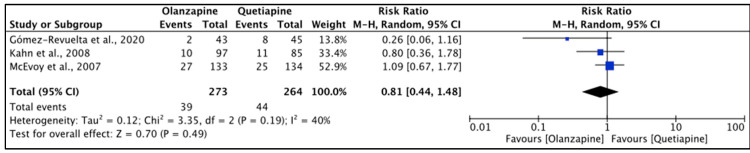
Long-term treatment akathisia in olanzapine vs. quetiapine RE [28,36,37].

**Figure 9 pharmacy-11-00175-f009:**
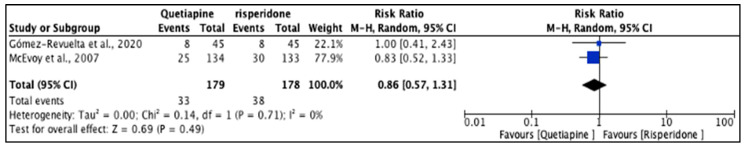
Long-term treatment-emergent akathisia in quetiapine vs. risperidone, RE [28,36].

**Figure 10 pharmacy-11-00175-f010:**
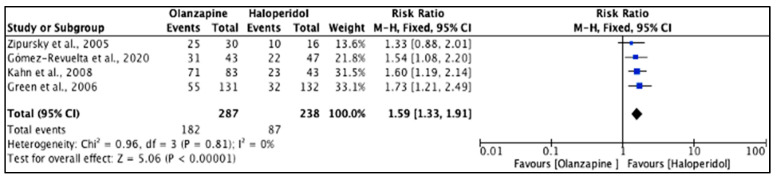
Weight gain for olanzapine vs. haloperidol long-term treatment—FE [33,36,37,39].

**Figure 11 pharmacy-11-00175-f011:**
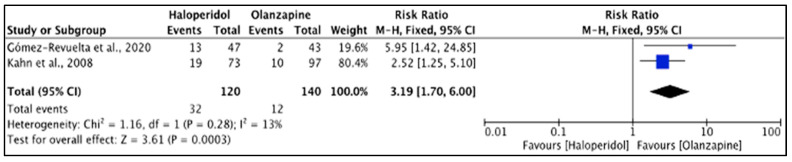
Long-term treatment akathisia in haloperidol vs. olanzapine in FE [36,37].

**Figure 12 pharmacy-11-00175-f012:**
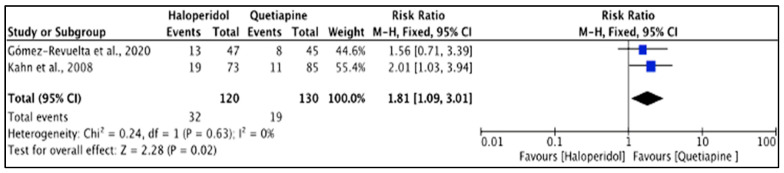
Long-term treatment akathisia in haloperidol vs. quetiapine FE [36,37].

**Figure 13 pharmacy-11-00175-f013:**
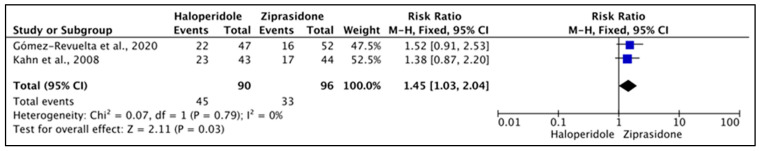
Weight gain for haloperidol vs. ziprasidone long-term treatment—FE [36,37].

**Figure 14 pharmacy-11-00175-f014:**
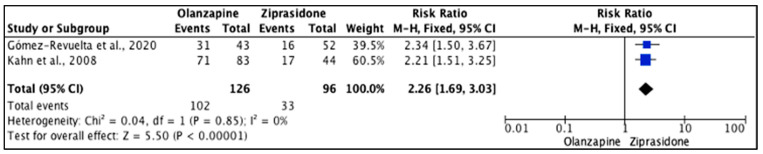
Weight gain for olanzapine vs. ziprasidone long-term treatment—FE [36,37].

**Figure 15 pharmacy-11-00175-f015:**
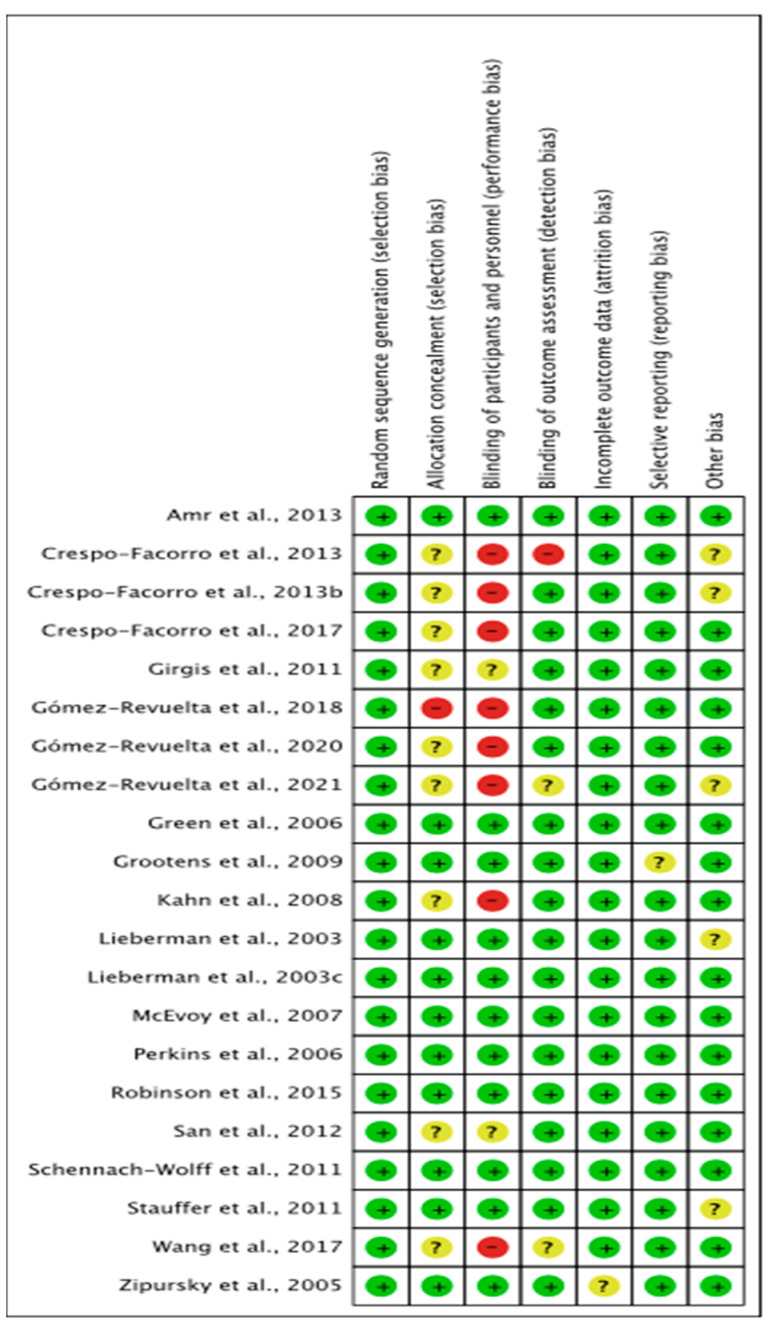
Traffic light risk of bias calculation [24,25,26,27,28,29,30,31,32,33,34,35,36,37,38,39,40,42,43,44].

**Table 1 pharmacy-11-00175-t001:** Definition and descriptions of scales used in the systematic review.

Scale	Parameters Measured	Range
Global Assessment of Functioning Scale (GAF)	Measures patient’s overall degree of impairment in psychosocial, occupational or educational functioning.	0–100, lower score is favourable
Simpson-Angus Scale (SAS or SARS)	Measures the severity of drug-induced parkinsonism symptoms.	0–40, lower score is favourable
Positive and Negative Syndrome Scale (PANSS)	Measures increasing levels of psychopathology.	30–210, lower score is favourable
Brief Psychiatric Rating Scale (BPRS)	Measures levels of psychopathology; there are more numbers of BPRS versions in use featuring 16 to 24 items.	1–7 per question, lower score is favourable
Scale for the Assessment of Positive Symptoms (SAPS)	Measures levels of positive symptoms of schizophrenia (delusions, hallucination and thought disorder).	0–170, lower score is favourable
Scale for the Assessment of Negative Symptoms (SANS)	Measures the levels of negative symptoms of schizophrenia (affective flattening or blunting, alogia, avolition–apathy, anhedonia–asociality, attention).	0–125, lower score is favourable
Calgary Depression Scale for Schizophrenia (CDSS)	Measures the level of depression in people with schizophrenia.	0–27, lower score is favourable
Young Mania Rating Scale (YMRS)	Measures manic symptoms (elevated mood, increased motor activity–energy, sexual interest, sleep, irritability, speech, language–thought disorder, content, disruptive–aggressive behaviour, appearance, insight).	0–60, lower score is favourable
Clinical Global Impression (CGI) Scale	Measures illness severity (CGIS), global improvement, change (CGIC) and therapeutic response.	3–21, lower score is favourable
Udvalg for Kliniske Undersogelser (UKU)	Measures the levels of psychotics’, neurological, autonomic and other side effects of antipsychotic drugs. There are two versions of UKU, one for patients (UKU-SERS-Pat) and one for clinicians (UKU-SERS-Clin).	0–120, lower score is favourable
Barnes Akathisia Scale (BAS or BARS)	Measures the severity of drug-induced akathisia, based on brief observation by the clinician of the patient.	0–20, lower score is favourable
Abnormal Involuntary Movement Scale (AIMS)	Measures the severity of dyskinesias (facial and oral movement, extremity movements, trunk movements, global movement and dental status, and a total severity score for abnormal movements).	0–28, lower score is favourable
St. Hans Rating Scale (SHRS)	Measures neuroleptic-induced hyperkinesia, parkinsonism symptoms, akathisia and dystonia.	0–48, lower score is favourable
Heinrichs–Carpenter Quality of Life Scale (HCQoL)	Measures schizophrenic deficit syndrome.	0–126, lower score is favourable

**Table 2 pharmacy-11-00175-t002:** Summary of plotted mean scores for aripiprazole vs. ziprasidone [33,34,35].

Scale/Range *	Aripiprazole	Ziprasidone	*p*-Values	Mean Difference	Order of Impact on Response to Therapy
CGI3–21	2.5 −4.2	2.3 −4.2	0.11 0.95	0.26 (−0.06, 0.58) −0.01 (−0.35, 0.32)	Ziprasidone Equal
BPRS 24–168	30.1 −37.6	29.7 −32.1	0.65 **<0.001**	0.45 (−1.48, 2.39) −5.46 (−8.37, −2.54)	Ziprasidone Ziprasidone
SANS 0–125	3.9 −2.8	3.2 −2.1	0.16 0.33	0.71 (−0.29, 1.71) −0.74 (−2.25, 0.76)	Ziprasidone Ziprasidone
SAPS 0–170	1.0 −13.5	1.1 −12.7	0.74 0.15	−0.10 (−0.70, 0.50) −0.74 (−1.73, 0.26)	Aripiprazole Aripiprazole
CDSS 0–27	1.5 −2.1	0.5 −1.6	**0.04** 0.25	0.87 (0.03, 1.71) −0.48 (−1.29, 0.34)	Ziprasidone Ziprasidone
YMRS 0–60	1.1 −10.7	1.6 −10.0	0.10 0.79	−0.54 (−1.20, 0.11) −0.68 (−5.72, 4.37)	Aripiprazole Aripiprazole

* Lower scale score is favourable; first value (columns 3 and 4): all studies average mean improvement endpoint, second value (columns 3 and 4): all studies average mean change from baseline, column 5: *p*-values (all studies average mean improvement at endpoint and mean change from baseline to endpoint) and column 6: drug with possible higher impact on the patient response (first line for mean score at endpoint, second line is for mean change between baseline and endpoint).

**Table 3 pharmacy-11-00175-t003:** Summary of plotted mean scores for aripiprazole vs. quetiapine [33,34,35].

Scale/Range *	Aripiprazole	Quetiapine	*p*-Values	Mean Difference	Order of Impact on Response to Therapy
CGI 3–21	2.5 −4.2	2.7 −3.9	0.32 0.10	−0.19 (−0.56, 0.18) −0.32 (−0.71, 0.07)	Aripiprazole Aripiprazole
BPRS 24–168	30.1 −37.6	31.1 −32.8	0.40 **0.002**	−0.93 (−3.08, 1.22) −4.77 (−7.83, −1.71)	Aripiprazole Aripiprazole
SANS 0–125	3.9 −2.8	3.9 −1.8	0.97 0.13	−0.02 (−1.21, 1.17) −1.06 (−2.43, 0.32)	Equal Aripiprazole
SAPS 0–170	1.0 −13.5	1.5 −9.6	0.09 0.32	−0.54 (−1.17, 0.09) −0.32 (−0.93, 0.30)	Aripiprazole Aripiprazole
CDSS 0–27	1.5 −2.1	1.3 −2.4	0.08 0.55	0.42 (−0.05, 0.89) 0.25 (−0.56, 1.06)	Quetiapine Quetiapine
YMRS 0–60	1.1 −10.7	1.9 −10.3	**0.01** 0.53	−0.82 (−1.46, −0.18) −0.43 (−1.78, 0.91)	Aripiprazole Aripiprazole

* Refer to Table 2 legend.

**Table 4 pharmacy-11-00175-t004:** Summary of plotted mean scores for ziprasidone vs. quetiapine [33,34,35].

Scale/Range *	Ziprasidone	Quetiapine	*p*-Values	Mean Difference	Order of Impact on Response to Therapy
CGI 3–21	2.3 −4.1	2.7 −3.9	**0.02** 0.28	−0.44 (−0.80, −0.77) −0.21 (−0.60, 0.17)	Ziprasidone Ziprasidone
BPRS 24–168	29.7 −32.1	31.0 −33.1	0.19 0.54	−1.38 (−3.45, 0.68) 1.17 (−2.60, 4.93)	Ziprasidone Quetiapine
SANS 0–125	3.2 −2.0	3.9 −1.8	0.21 0.77	−0.74 (−1.89, 0.41) −0.20 (−1.55, 1.14)	Ziprasidone Ziprasidone
SAPS 0–170	1.1 −12.7	1.5 −12.4	0.27 0.57	−0.40 (−1.10, 0.31) −0.33 (−1.49, 0.82)	Ziprasidone Quetiapine
CDSS 0–27	0.5 −1.6	0.3 −2.4	**0.03** 0.06	0.20 (0.02, 0.38) 0.73 (−0.04, 1.49)	Quetiapine Quetiapine
YMRS 0–60	1.6 −10.0	3.4 −10.3	0.52 0.70	−0.26 (−1.07, −0.54) 0.27 (−1.10, 1.63)	Ziprasidone Quetiapine

* Refer to Table 2 legend.

**Table 5 pharmacy-11-00175-t005:** Summary of plotted RR for aripiprazole vs. risperidone [31,35].

Side Effects *	Aripiprazole	Risperidone	*p*-Values	RR (95% CI)	Impact
Concentration difficulties	15%	13.8%	0.60	1.28 (0.51, 3.25)	High
Increased fatigability	33.6%	36.6%	0.48	0.91 (0.72, 1.17)	Low
Increased duration of sleep	8.1%	11.6%	0.21	0.70 (0.40, 1.23)	Low
Akinesia	9.8%	8.0%	0.47	1.24 (0.69, 2.22)	High
Weight gain	18.3%	16.1%	0.53	1.13 (0.77, 1.67)	High
Diminished sexual desire	4.7%	12.5%	**0.01**	0.38 (0.18, 0.82)	Low
Constipation	4.3%	5.4%	0.58	0.79 (0.35, 1.78)	Low

* Columns 3 and 4: Prevalence of events; column 5: RR (95% CI)—when the estimated effect size or point estimate is >1, it indicates high impact, =1 indicates no impact, <1 indicates low impact; and column 6: Drug type impact on the patient experience of side effects.

**Table 6 pharmacy-11-00175-t006:** Summary of plotted RR for aripiprazole vs. ziprasidone [35,40,41].

Side Effects	Aripiprazole	Ziprasidone	*p*-Values	RR (95% CI) *	Impact
Concentration difficulties	15.5%	15.6%	0.95	0.98 (0.61, 1.59)	Low
Increased fatigability	29.5%	35%	0.22	0.83 (0.63, 1.11)	Low
Sleepiness	29.5%	39.4%	**0.03**	0.74 (0.57, 0.97)	Low
Memory impairment	4.6%	5.8%	0.67	0.79 (0.26, 2.37)	Low
Depression	4.5%	5.6%	0.63	0.80 (0.33, 1.97)	Low
Restlessness	4.6%	5.8%	0.67	0.79 (0.26, 2.37)	Low
Increased duration of sleep	16%	28.8%	**0.003**	0.55 (0.37, 0.82)	Low
Rigidity	3%	11.5%	**0.02**	0.26 (0.09, 0.79)	Low
Akinesia	28.8%	30%	0.86	0.97 (0.65, 1.44)	Low
Tremors	13.6%	5.8%	0.06	2.36 (0.97, 5.74)	High
Increased salivation	12%	11.3%	0.87	1.05 (0.60, 1.82)	High
Constipation	7.5%	7%	0.84	1.08 (0.51, 2.26)	High
Vertigo	4.6%	7.7%	0.32	0.59 (0.21, 1.65)	Low
Amenorrhea	5.5%	10.6%	0.87	0.51 (0.25, 1.06)	Low
Galactorrhoea	0%	4.6%	0.06	0.13 (0.02, 1.11)	Low
Diminished sexual desire	6.5%	7%	0.87	0.94 (0.43, 2.04)	Low
Orgasmic dysfunction	1.5%	5.8%	0.10	0.26 (0.05, 1.27)	Low
Erectile dysfunction	2%	9.4%	**0.005**	0.23 (0.08, 0.65)	Low
Ejaculatory dysfunction	2%	7.5%	**0.02**	0.29 (0.10, 0.82)	Low
Weight gain	39.5%	26.3%	**0.01**	1.50 (1.10, 2.03)	High

* Refer to Table 5 legend.

**Table 7 pharmacy-11-00175-t007:** Summary of plotted RR for aripiprazole vs. quetiapine [35,40,41].

Side Effects	Aripiprazole	Quetiapine	*p*-Values	RR (95% CI)	Impact
Concentration difficulties	11.5%	10%	0.82	1.10 (0.49, 2.48)	High
Increased fatigability	29.5%	37%	0.06	0.76 (0.58, 1.01)	Low
Sleepiness	27.5%	44.7%	**<0.001**	0.58 (0.45, 0.75)	Low
Depression	4.5%	2.1%	0.26	2.11 (0.58, 7.66)	High
Restlessness	4.6%	2.2%	0.37	2.05 (0.42, 9.91)	High
Increased duration of sleep	16%	29.1%	**0.001**	0.53 (0.36, 0.78)	Low
Rigidity	3%	2.2%	0.72	1.36 (0.6, 7.29)	High
Tremors	13.7%	4.4%	**0.04**	3.07 (1.07, 8.77)	High
Increased salivation	10%	13.5%	0.36	0.72 (0.35, 1.47)	Low
Constipation	7.5%	12.8%	0.13	0.60 (0.31, 1.17)	Low
Vertigo	4.6%	0%	0.14	4.81 (0.60, 38.41)	High
Amenorrhea	5.5%	0%	0.06	5.33 (0.96, 29.48)	High
Galactorrhoea	0%	4.3%	**0.03**	0.10 (0.01, 0.78)	Low
Diminished sexual desire	6.5%	10.6%	0.18	0.62 (0.30, 1.26)	Low
Orgasmic dysfunction	1.5%	4.4%	0.21	0.34 (0.06, 1.82)	Low
Erectile dysfunction	2%	10.6%	**0.002**	0.20 (0.07, 0.56)	Low
Ejaculatory dysfunction	5.5%	10%	0.42	0.52 (0.10, 2.58)	Low
Weight gain	39.5%	35.5%	0.49	1.10 (0.84, 1.44)	High
Memory impairment	4.6%	2.2%	0.37	2.05 (0.42, 9.91)	High

**Table 8 pharmacy-11-00175-t008:** Summary of plotted RR for quetiapine vs. ziprasidone [35,40,41].

Side Effects	Quetiapine	Ziprasidone	*p*-Values	RR (95% CI)	Impact
Concentration difficulties	10.6%	17.5%	0.11	0.62 (0.35, 1.11)	Low
Increased fatigability	37%	33.1%	0.35	1.14 (0.86, 1.50)	High
Sleepiness	44.7%	38.1%	0.07	1.22 (0.98, 1.52)	High
Memory impairment	2.2%	5.8%	0.24	0.39 (0.81, 1.86)	Low
Depression	2.1%	2.6%	0.14	0.38 (0.10, 1.37)	Low
Restlessness	2.2%	5.8%	0.24	0.39 (0.08, 1.86)	Low
Increased duration of sleep	29.1%	28.8%	0.79	1.05 (0.71, 1.56)	High
Rigidity	2.2%	12.5%	**0.03**	0.19 (0.04, 0.84)	Low
Tremors	4.4%	5.8%	0.68	0.77 (0.22, 2.64)	Low
Increased salivation	13.5%	8.3%	0.44	2.21 (0.29, 16.63)	High
Constipation	12.8%	7%	0.11	1.81 (0.58, 3.70)	High
Vertigo	0%	7.7%	**0.05**	0.13 (0.02, 0.99)	Low
Amenorrhea	0%	10.6%	**0.006**	1.10 (0.02, 0.50)	High
Galactorrhoea	4.3%	4.4%	0.81	1.16 (0.34, 4.02)	High
Diminished sexual desire	10.6%	7%	0.24	1.56 (0.74, 3.29)	High
Orgasmic dysfunction	4.4%	5.8%	0.68	0.77 (0.22, 2.64)	Low
Erectile dysfunction	10.6%	9.4%	0.75	1.17 (0.45, 3.00)	High
Ejaculatory dysfunction	10%	11.3%	0.84	0.91 (0.35, 2.35)	Low
Weight gain	44%	29%	**0.003**	1.49 (1.15, 1.94)	High

**Table 9 pharmacy-11-00175-t009:** Summary of plotted RR for olanzapine vs. quetiapine [28,37,38].

Side Effects	Olanzapine	Quetiapine	*p*-Values	RR (95% CI)	Impact
Weight gain	65.6%	47%	**<0.001**	1.34 (1.15, 1.56)	High
Sleepiness	54.6%	60.3%	0.26	0.90 (0.75, 1.08)	Low
Increased duration of sleep	33.5%	42%	0.11	0.80 (0.61, 1.05)	Low
Akinesia	19.3%	23%	0.45	0.62 (0.18, 2.14)	Low
Constipation	9.1%	13%	0.26	0.71 (0.39, 1.30)	Low
Galactorrhoea	2.3%	0%	0.15	4.84 (0.56, 42.07)	High
Diminished sexual desire	23.3%	23%	0.91	1.02 (0.70, 1.49)	High
Orgasmic dysfunction	13.6%	13%	0.84	1.05 (0.62, 1.79)	High

**Table 10 pharmacy-11-00175-t010:** Summary of plotted RR for risperidone vs. quetiapine [28,36].

Side Effects	Risperidone	Quetiapine	*p*-Values	RR (95% CI)	Impact
Weight gain	45.5%	41.3%	0.39	1.11 (0.88, 1.41)	High
Sleepiness	47.8%	60.3%	0.10	0.76 (0.55, 1.05)	Low
Increased duration of sleep	24.2%	42%	**0.02**	0.54 (0.32, 0.91)	Low
Akinesia	24.2%	23%	0.76	1.06 (0.73, 1.54)	High
Constipation	12.4%	14.6%	0.57	0.86 (0.50, 1.46)	Low
Galactorrhoea	2.8%	0%	0.10	5.96 (0.72, 49.02)	High
Orgasmic dysfunction	15.7%	13%	0.44	1.22 (0.74, 2.03)	High

**Table 11 pharmacy-11-00175-t011:** Summary of plotted RR for olanzapine vs. risperidone [29,35].

Side Effects	Olanzapine	Risperidone	*p*-Values	RR (95% CI)	Impact
Weight gain	56.3%	45.5%	**0.03**	1.24 (1.02, 1.52)	Low
Sleepiness	54.6%	47.8%	0.21	1.14 (0.93, 1.40)	High
Increased duration of sleep	33.5%	24.2%	0.12	1.42 (0.92, 2.20)	Low
Akinesia	19.3%	24.7%	0.37	0.60 (0.19, 1.86)	Low
Constipation	9.1%	12.4%	0.39	0.77 (0.39, 1.44)	Low
Galactorrhoea	2.3%	2.8%	0,77	0.82 (0.22, 3.05)	Low
Diminished sexual desire	23.3%	24.2%	0.88	0.97 (0.67, 1.41)	Low
Orgasmic dysfunction	13.6%	15.7%	0.56	0.86 (0.52, 1.42)	Low

**Table 12 pharmacy-11-00175-t012:** Summary of plotted RR for quetiapine vs. olanzapine discontinuation by reasons [25,28,36,37].

Side Effects *	Quetiapine	Olanzapine	*p*-Values	RR (95% CI)	Impact
Lack of efficacy	25.7%	7.2%	<0.001	3.48 (3.05, 5.91)	High
Side effects	7.4%	9.1%	0.44	0.81 (0.49, 1.37)	Low
Drop out	21.2%	12.5%	0.50	2.55 (0.17, 38.18)	Low
Lack of compliance	14%	17.5%	0.30	0.77 (0.47, 1.27)	Low
Others	26.3%	24.4%	0.62	1.06 (0.83, 1.36)	Low

* Columns 3 and 4: Prevalence of events; column 5: RR (95% CI)—when the estimated effect size or point estimate is >1, it indicates high impact, =1 indicates no impact, <1 indicates low impact; and column 6: Drug type impact on the patient experience of side effects.

**Table 13 pharmacy-11-00175-t013:** Summary of plotted RR for quetiapine vs. risperidone discontinuation by reasons [26,29,35,36].

Side Effects	Quetiapine	Risperidone	*p*-Values	RR (95% CI)	Impact
Lack of efficacy	36.5%	17%	0.85	0.68 (0.01, 33.70)	Low
Side effects	10.6%	24%	0.02	0.44 (0.21, 0.89)	Low
Drop out	21.2	21.6	0.91	1.03 (0.66, 1.78)	High

**Table 14 pharmacy-11-00175-t014:** Summary of plotted RR for quetiapine vs. ziprasidone discontinuation by reasons [25,35,36,37,39,41].

Side Effects	Quetiapine	Ziprasidone	*p*-Values	RR (95% CI)	Impact
Lack of efficacy	39.6%	19.4%	<0.001	2.05 (1.57, 2.68)	High
Side effects	8.3%	26%	<0.001	0.34 (0.22, 0.51)	Low
Drop out	15.5%	10.8%	0.08	1.47 (0.95, 2.27)	High
Lack of compliance	16.3%	11.2%	0.15	1.41 (0.88, 2.27)	High
Others	2.4%	2%	0.84	1.18 (0.24, 5.72)	High

**Table 15 pharmacy-11-00175-t015:** Summary of plotted RR for quetiapine vs. haloperidol discontinuation by reasons [25,36,37].

Side Effects	Quetiapine	Haloperidol	*p*-Values	RR (95% CI)	Impact
Lack of efficacy	35.5%	29%	0.80	1.14 (0.41, 3.16)	High
Side effects	5.8%	12.8%	0.01	0.43 (0.22, 0.84)	Low
Drop out	14%	19.5%	0.15	0.70 (0.43, 1.14)	Low
Lack of compliance	21.2%	17%	0.50	1.23 (0.68, 2.23)	High
Others	2.4%	3.3%	0.62	0.70 (0.16, 2.93)	Low

**Table 16 pharmacy-11-00175-t016:** Summary of plotted RR for quetiapine vs. aripiprazole discontinuation by reasons [25,36,37].

Side Effects	Quetiapine	Aripiprazole	*p*-Values	RR (95% CI)	Impact
Lack of efficacy	47.3%	12.8%	<0.001	3.69 (2.56, 5.32)	High
Side effects	12.4%	12%	0.90	1.03 (0.62, 1.73)	High
Drop out	18.3%	27%	0.04	0.68 (0.47, 0.98)	Low
Lack of compliance	13%	10.3%	0.40	1.26 (0.74, 2.14)	High

**Table 17 pharmacy-11-00175-t017:** Summary of plotted RR for aripiprazole vs. ziprasidone discontinuation by reasons [35,36,39].

Side Effects	Aripiprazole	Ziprasidone	*p*-Values	RR (95% CI)	Impact
Lack of efficacy	14.1%	20%	0.11	0.71 (0.46, 1.09)	Low
Side effects	12%	34.4%	<0.001	0.34 (0.23, 0.52)	Low
Lack of compliance	27%	12.4%	0.0005	2.18 (1.41, 3.37)	High

**Table 18 pharmacy-11-00175-t018:** Summary of plotted RR for risperidone vs. olanzapine discontinuation by reasons [25,28,36].

Side Effects	Risperidone	Olanzapine	*p*-Values	RR (95% CI)	Impact
Lack of efficacy	12.2%	10.8%	0.64	1.17 (0.61, 2.22)	High
Side effects	14.4%	11.3%	0.49	1.29 (0.62, 2.68)	High

**Table 19 pharmacy-11-00175-t019:** Summary of plotted RR for risperidone vs. ziprasidone discontinuation by reasons [25,36].

Side Effects	Risperidone	Ziprasidone	*p*-Values	RR (95% CI)	Impact
Lack of efficacy	17.1%	18.3%	0.90	1.08 (0.33, 3.58)	High
Side effects	24%	33%	0.42	0.60 (0.17, 2.09)	Low
Drop out	21.6%	16%	0.59	1.24 (0.57, 2.67)	High

**Table 20 pharmacy-11-00175-t020:** Summary of plotted RR for risperidone vs. haloperidol discontinuation by reasons [25,36].

Side Effects	Risperidone	Haloperidol	*p*-Values	RR (95% CI)	Impact
Lack of efficacy	17.1%	23.4%	0.30	0.72 (0.39, 1.33)	Low
Side effects	27.3%	26%	0.81	1.06 (0.65, 1.73)	High
Drop out	21.6%	17%	0.46	1.26 (0.68, 2.32)	High

**Table 21 pharmacy-11-00175-t021:** Summary of plotted RR for olanzapine vs. ziprasidone discontinuation by reasons [25,36,37].

Side Effects	Olanzapine	Ziprasidone	*p*-Values	RR (95% CI)	Impact
Lack of efficacy	9.7%	19.5%	0.01	0.50 (0.29, 0.86)	Low
Side effects	8.1%	26.8%	<0.001	0.33 (0.19, 0.57)	Low
Lack of compliance	17.5%	10.4%	0.05	1.79 (1.00, 3.20)	High
Drop out	17.5%	17.1%	0.93	0.94 (0.26, 3.39)	Low
Others	1.5%	2%	0.75	0.74 (0.11, 4.85)	Low

**Table 22 pharmacy-11-00175-t022:** Summary of plotted RR for olanzapine vs. haloperidol discontinuation by reasons [25,27,33,36,37,38].

Side Effects	Olanzapine	Haloperidol	*p*-Values	RR (95% CI)	Impact
Lack of efficacy	9.8%	22%	<0.001	0.47 (0.32, 0.68)	Low
Side effects	7.2%	15.7%	0.001	0.47 (0.29, 0.75)	Low
Lack of compliance	16.2%	20.6%	0.18	0.79 (0.56, 1.12)	High
Drop out	17.5%	17%	0.96	0.97 (0.28, 3.31)	Low
Others	23.7%	23%	0.64	1.06 (0.84, 1.34)	Low

**Table 23 pharmacy-11-00175-t023:** Summary of plotted RR for haloperidol vs. ziprasidone discontinuation by reasons [25,36,37].

Side Effects	Haloperidol	Ziprasidone	*p*-Values	RR (95% CI)	Impact
Lack of efficacy	29%	19.5%	0.06	1.46 (0.98, 2.15)	High
Side effects	17.2%	11.6%	0.10	1.55 (0.92, 2.63)	High
Lack of compliance	19.5%	9.7%	0.01	2.11 (1.17, 3.79)	High
Drop out	17%	16%	0.98	1.01 (0.52, 1.94)	High
Others	1.6%	2%	0.89	0.89 (0.15, 5.07)	Low

## Data Availability

All generated data is provided in the Appendix A, any additional clarifications can be provided by contacting the corresponding author.

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
