# Peer review of "Efficacy and Tolerance of Antipsychotics Used for the Treatment of Patients Newly Diagnosed with Schizophrenia: A Systematic Review and Meta-Analysis"

_pharmacy, 2023, doi:10.3390/pharmacy11060175_

Round 1

Reviewer 1 Report

Comments and Suggestions for Authors

How the quality of included study was assessed ?

What were inclusion and exclusion criteria ?

Author Response

Thank you, we are native English speakers and reviewed the paper thoroughly 

Reviewer 2 Report

Comments and Suggestions for Authors

The manuscript by Morrissey and cols represents a well-written review of the literature on antipsychotics for schizophrenia management. Documents used in the review well selected and rigorous scientific interpreted, thus validating adequate review statements. I feel that it will interest the researchers in the area.

I have only few comments:

1. Review the authors. The ORCID of an author does not present in the authorship list appears. Nilamadhab Kar: https://orcid.org/0000-0002-8801-9245

2. Following the journal's instructions: The summary must have a total of a maximum of 200 words. Please adjust it to the recommendations.

3. Line 116. Aim 3. Why did you decide to collect information from India and the UK? What is the reason for choosing these countries? This objective is not justified.

4. Point 4 of Search Strategy, I would mention as point 3.1 within the methodology. And the same with point 5, I would categorize it as point 3.2.

5. Line 181, define the acronyms ZQ and HM.

6. The results are a little tedious to read, as they refer to so many authors, which makes the thread of the result of the meta-analysis a little lost.

7. In the discussion section I would add the discussion of each drug. For example: 9. Discussion 9.1 Olanzapine, 9.2 Clozapine, 9.3 Haloperidol, etc..

Reviewer 3 Report

Comments and Suggestions for Authors

This is a systematic review that includes a comparative analysis of the efficacy and tolerance of antipsychotic drugs based on the existing literature.  The comparative analysis of continuous and dichotomous outcomes from patient surveys makes this study quite comprehensive. However, the presentation of the findings was underwhelming, and the study lacked impact. Moreover, there were a few issues with the writing. Overall, the manuscript needs a major revision. Following are my specific suggestions/comments to the authors.

1)    The abstract is full of blanket statements like “Aripiprazole was more tolerated than risperidone, ziprasidone, and quetiapine” (line 26). The authors mention the specific variables in question after the statements. for e,g, “…on diminished sexual desire, sleepiness, increased duration of sleep, rigidity, erectile dysfunction, ejaculatory dysfunction, tremor, and akathisia events, p < 0.05...” (line 27). This style of presentation is repeated throughout the abstract and some parts of the manuscript and makes comprehension of the results quite confusing. I suggest rewording these statements to minimize ambiguity.

2)    Some results presented in the manuscript are counterintuitive. One prominent example is in lines 38-40: “Ziprasidone caused less discontinuation than quetiapine, aripiprazole, and haloperidol (lack of efficacy and lack of compliance (p < 0.05). Aripiprazole caused less discontinuation than quetiapine, ziprasidone, and olanzapine (lack of efficacy and side effects (p < 0.05).”  After reading these contradictory statements, I do not know whether Ziprasidone or Aripiprazole had less discontinuation due to lack of efficacy. This may even call into question the robustness of the comparative statistics, and whether the authors should only focus on one parameter e.g. lack of efficacy, for their statistical comparisons.

3)    The authors show robust statistical comparisons showing which antipsychotic drug was better tolerated or preferred for Akathisia and weight gain as side effects. These results can potentially be exciting yet are not highlighted as key findings. Please consider including these findings in the conclusions and as a part of the core message. I feel similar comparisons for other major side effects of antipsychotic drugs is also warranted.

4)    Long and short-term exposure to antipsychotics can lead to cardiovascular side effects, that can be potentially life-threatening.  This review did not explore these side effects. Please discuss why cardiovascular side effects were not included.

5)    In Table 1, which includes scales that were administered and their respective ranges, authors should also include what parameters these scale measures. The authors should describe the consistency with which these scales were administered in the methodology section. Were all scales administrated for every volunteer? Or were the scales consistent when assessing the same drug? If there were two scales with different ranges, assessing the same variable, the authors should describe if and how they normalized the two scores.  

6)    Were there instances where authors had to compare or combine continuous and dichotomous data? If yes, then please describe the methodology.

7)    The abbreviations like APD, RCTs, FGA, and EPS are not defined in the text.

8)    There are tense agreement issues throughout the manuscript with punctuation marks missing in several places. Worlds in figure/table captions are capitalized seemingly at random.

Comments on the Quality of English Language

There are tense agreement issues throughout the manuscript with punctuation marks missing in several places. Worlds in figure/table captions are capitalized seemingly at random.

Reviewer 4 Report

Comments and Suggestions for Authors

The authors present a systematic review and meta-analysis of antipsychotic efficacy and side effects. The systematic review appears well done and thorough. One potential issue with the manuscript is due to how large the review is and the summarization of the findings. The manuscript results are almost overwhelming to comprehend and could benefit from some attempt to create a more concise summary of the findings while utilizing the supplementary information for lengthy tables and detailed findings. The discussion does attempt to do this with the "summary of findings" portion of each sub-topic but at this point the reader is 26 pages into the manuscript.

-The abstract, as currently written is too long and just lists finding after finding. Consider shortening the abstract by describing the top findings.

-Lines 86-90, although its fine to highlight the prolactinemia and sexual dysfunction by antipsychotics, the primary differentiating factor between the classes should be highlighted which is their receptor profiles. Additionally, most psychiatrists would suggest the movement side effects versus metabolic side effects are the primary features of 1st (typical) versus 2nd (atypical) generation antipsychotics. Perhaps mentioning some of these primary differences could be considered?

-Something that should be laid out along with your review aim or objective, is the background that includes what other systematic reviews/meta-analyses have been performed on your topic and why your review is needed. Some rationale would be helpful.

-Please provide the rationale for time-limiting your systematic review

-how was screening performed? In a software program? In excel?

-In the results, is there a way to give a summarizing statement at the beginning of each section. The results is written well but is just a listing of each studies finding which makes it difficult to comprehend especially given the number of studies included in your review and the length. Some sort of summarizing feature would be great here.

-I wonder if there is a way to make your tables more concise in the main manuscript such as creating a table that includes all of the head-to-head comparisons with a + or - (or some other symbol) that indicates the findings. These tables are great and should be kept in the supplementary material if such a change is made.

Author Response

Please see attached

The authors present a systematic review and meta-analysis of antipsychotic efficacy and side effects. The systematic review appears well done and thorough. One potential issue with the manuscript is due to how large the review is and the summarization of the findings. The manuscript results are almost overwhelming to comprehend and could benefit from some attempt to create a more concise summary of the findings while utilizing the supplementary information for lengthy tables and detailed findings.

Thank you.

The discussion does attempt to do this with the "summary of findings" portion of each sub-topic but at this point the reader is 26 pages into the manuscript.

Corrected, please see comments to reviewer 2 and 3

-The abstract, as currently written is too long and just lists finding after finding. Consider shortening the abstract by describing the top findings.

Corrected

-Lines 86-90, although its fine to highlight the prolactinemia and sexual dysfunction by antipsychotics, the primary differentiating factor between the classes should be highlighted which is their receptor profiles. Additionally, most psychiatrists would suggest the movement side effects versus metabolic side effects are the primary features of 1st (typical) versus 2nd (atypical) generation antipsychotics. Perhaps mentioning some of these primary differences could be considered?

Corrected

-Something that should be laid out along with your review aim or objective, is the background that includes what other systematic reviews/meta-analyses have been performed on your topic and why your review is needed. Some rationale would be helpful.

Corrected – please see discussion.

-Please provide the rationale for time-limiting your systematic review

Corrected – please see the selection criteria.

-how was screening performed? In a software program? In excel?

This was included and removed based on the other reviewers comments but we are happy to add as supp.

-In the results, is there a way to give a summarizing statement at the beginning of each section. The results is written well but is just a listing of each studies finding which makes it difficult to comprehend especially given the number of studies included in your review and the length. Some sort of summarizing feature would be great here.

Corrected

-I wonder if there is a way to make your tables more concise in the main manuscript such as creating a table that includes all of the head-to-head comparisons with a + or - (or some other symbol) that indicates the findings. These tables are great and should be kept in the supplementary material if such a change is made.

Corrected

Round 2

Reviewer 3 Report

Comments and Suggestions for Authors

The authors have addressed all of my concerns from the initial review. However, I do have a couple of minor comments.

1)      Cardiovascular side effects such as postural hypotension and tachycardia are quite prevalent with the usage of antipsychotics and can manifest even after short-term use. In my initial review, I recommended that the authors should discuss why cardiovascular side effects were not included. In their response, the authors state that cardiovascular side effects were implied under metabolic changes, and hypertension was occasionally reported in their selected pool of literature and included when it was possible. Firstly, these side effects occur as a direct effect on ion channels and α2-adrenoceptors (both cardiac and peripheral nervous system) and as far as I know, have no relation to blood glucose levels. Secondly, I could not find a single mention of hypertension in the manuscript.

I urge the authors to discuss the exclusion of cardiovascular side effects in the manuscript, or a clarification stating that the review is limited to non-cardiovascular extrapyramidal symptoms.

2)      Authors state the selection criteria for the literature, and the keywords used for database searches. However, Figure 1 suggests after using “refined keywords” with recent dates, the selected articles dropped from 14417 to 166. I recommend that the authors should clarify which of the keywords were used for the initial search and which ones were “refined”.

Comments on the Quality of English Language

Minor tense agreement issues are still present in the current format. I would also like to point out that the revised manuscript uploaded in response is showing the "track changes". This makes the review much difficult as the reviewers have to sift through deletions, to formulate a coherent sentence.

Author Response

1)      Cardiovascular side effects such as postural hypotension and tachycardia are quite prevalent with the usage of antipsychotics and can manifest even after short-term use. In my initial review, I recommended that the authors should discuss why cardiovascular side effects were not included. In their response, the authors state that cardiovascular side effects were implied under metabolic changes, and hypertension was occasionally reported in their selected pool of literature and included when it was possible. Firstly, these side effects occur as a direct effect on ion channels and α2-adrenoceptors (both cardiac and peripheral nervous system) and as far as I know, have no relation to blood glucose levels. Secondly, I could not find a single mention of hypertension in the manuscript.

I urge the authors to discuss the exclusion of cardiovascular side effects in the manuscript, or a clarification stating that the review is limited to non-cardiovascular extrapyramidal symptoms.

We covered the issue in the introduction. We did not exclude cardiovascular side effects, they were not reported in our selected studies, New paragraph added to explain

2)      Authors state the selection criteria for the literature, and the keywords used for database searches. However, Figure 1 suggests after using “refined keywords” with recent dates, the selected articles dropped from 14417 to 166. I recommend that the authors should clarify which of the keywords were used for the initial search and which ones were “refined”.

Corrected

Comments on the Quality of English Language

Minor tense agreement issues are still present in the current format. I would also like to point out that the revised manuscript uploaded in response is showing the "track changes". This makes the review much difficult as the reviewers have to sift through deletions, to formulate a coherent sentence.

Apology, but this is the journal requirement. Tense review was done again.

Thank you

Hana

Reviewer 4 Report

Comments and Suggestions for Authors

None

Author Response

Thank you